# CAE v2: Context Autoencoder with CLIP Latent Alignment

**Xinyu Zhang**[1][*], **Jiahui Chen**[2,1][*], **Junkun Yuan**[3,1], **Qiang Chen**[1], **Jian Wang**[1], **Xiaodi Wang**[1], **Shumin Han**[1], **Xiaokang Chen**[4,1], **Jimin Pi**[1], **Kun Yao**[1], **Junyu Han**[1], **Errui Ding**[1], **Jingdong Wang**[1][†]

*{zhangxinyu14,chenqiang13,wangjian33,wangxiaodi03,hanshuming,pijimin01,yaokun01,hanjunyu,dingerrui}@baidu.com,*
*wangjingdong@baidu.com, jiahui.chen@buaa.edu.cn, yuanjk@zju.edu.cn, pkucxk@pku.edu.cn*
[1] *Baidu VIS* [2] *School of Automation Science and Electrical Engineering, Beihang University*
[3] *College of Computer Science and Technology, Zhejiang University* [4] *Peking University*

**Reviewed on OpenReview:** *https://openreview.net/forum?id=f36LaK7MOF*

## Abstract

Masked image modeling (MIM) learns visual representations by predicting the masked patches on a pre-defined target. Inspired by MVP (Wei et al., 2022b) that displays impressive gains with CLIP, in this work, we also employ the semantically rich CLIP latent as target and further tap its potential by introducing a new MIM pipeline, CAE v2, to learn a high-quality encoder and facilitate model convergence on the pre-training task. CAE v2 is an improved variant of CAE (Chen et al., 2023), applying the CLIP latent on two pretraining tasks, *i.e.*, visible latent alignment and masked latent alignment. Visible latent alignment directly mimics the visible latent representations from the encoder to the corresponding CLIP latent, which is beneficial for facilitating model convergence and improving the representative ability of the encoder. Masked latent alignment predicts the representations of masked patches within the feature space of CLIP latent as standard MIM task does, effectively aligning the representations computed from the encoder and the regressor into the same domain. We pretrain CAE v2 on ImageNet-1K images and evaluate on various downstream vision tasks, including image classification, semantic segmentation, object detection and instance segmentation. Experiments show that our CAE v2 achieves competitive performance and even outperforms the CLIP vision encoder, demonstrating the effectiveness of our method. Code is available at `https://github.com/Atten4Vis/CAE`.

## 1 Introduction

Masked image modeling (MIM) (Bao et al., 2022) task has attracted numerous attention in self-supervised representation learning, showing strong performance on a variety of downstream tasks. Previous MIM methods (Bao et al., 2022; He et al., 2022; Xie et al., 2022; Chen et al., 2023) usually mask out some image patches, and then predict these masked patches conditioned on representations of visible patches according to specific prediction targets. The architecture of these MIM methods can be unified with an encoder-decoder format, in which the encoder is used for the representation learning and the decoder is used for the prediction of the masked patches. When transferring to downstream tasks, MIM only maintains the learned encoder and discards other parts.

A high-quality pre-trained encoder can greatly benefit downstream tasks. To improve the encoding quality, previous works take efforts on two aspects, *i.e.*, using better pretraining targets and decoupling the learning of encoder and decoder. As for the type of target, instead of low-level signals like RGB (He et al., 2022;

---

[*]Equal contribution.
[†]Corresponding author.
Work was done when J. Chen, J. Yuan and X. Chen were interns in Baidu VIS.

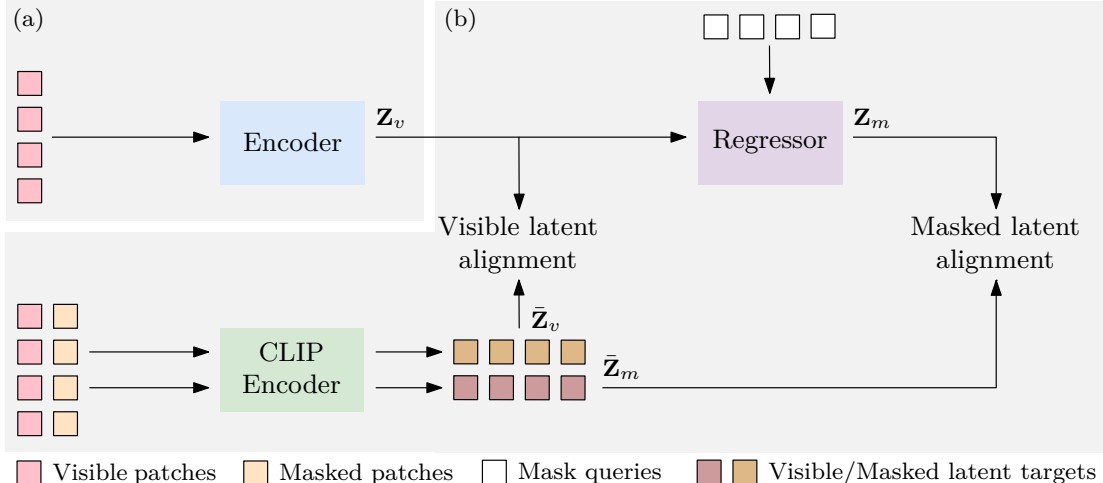

Figure 1: The pipeline of CAE v2. (a) The encoder receives the visible patches and extracts the visible latent representations $\mathbf{Z}_v$, which is then put into the regressor to generate masked latent representations $\mathbf{Z}_m$ conditioned on the mask queries. CAE v2 consists of two loss functions: visible latent alignment - mimicking $\mathbf{Z}_v$ to the visible latent targets $\bar{\mathbf{Z}}_v$, and masked latent alignment - regressing $\mathbf{Z}_m$ with the masked latent targets $\bar{\mathbf{Z}}_m$. Both $\bar{\mathbf{Z}}_v$ and $\bar{\mathbf{Z}}_m$ are generated by the vision branch of CLIP model. After pretraining, the encoder (a) is applied to downstream tasks, while (b) is replaced with the downstream task part.

Xie et al., 2022) and hand-crafted features like HOG (Wei et al., 2022a), some methods intuitively use high-level semantically rich targets to help the encoder learn more informative semantics, *e.g.*, discrete visual tokens (Bao et al., 2022; Chen et al., 2023; El-Nouby et al., 2021; Peng et al., 2022a; Dong et al., 2021), features from momentum encoders (Tao et al., 2022; Chen et al., 2022; Wu et al., 2023), and features from pretrained models (Wei et al., 2022b;a; Fang et al., 2023b;a; Hou et al., 2022; Peng et al., 2022b). Among these literature, MVP (Wei et al., 2022b) first utilizes CLIP latent as the masked prediction target in MIM, showing impressive gains on various downstream tasks. Another line like MAE (He et al., 2022) and CAE (Chen et al., 2023) attempts to partition the representation learning of the encoder and the reconstruction of the masked patches. For example, CAE introduces a latent contextual regressor to explicitly decouple the encoder learning and the decoder reconstruction, which effectively motivates encoder's power.

This paper also aims to acquire a high-quality vision encoder. Specifically, we introduce a new MIM pipeline, CAE v2, *i.e.*, a context autoencoder with CLIP latent alignment. The pretraining target of CAE v2 is CLIP latent following MVP (Wei et al., 2022b). The architecture of CAE v2 is built upon the CAE (Chen et al., 2023) method, with the variation on retaining the encoder and the regressor yet discarding the decoder. This modification results in a more lightweight model structure, enabling faster model pretraining and less computational cost.

CAE v2 contains two concurrent pretraining tasks, *i.e.*, visible latent alignment and masked latent alignment. Visible latent alignment is designed for an explicit optimization on the encoder, which imposes visible latent representations from the encoder to be close to those from the vision branch of CLIP model. It encourages the encoder to learn semantically rich information brought by supervision signals directly, which is beneficial for model convergence and performance improvement. Masked latent alignment is responsible for the masked patch prediction. It directly regresses feature representations of the masked patches from the regressor to the corresponding CLIP latents as standard MIM methods do. These two tasks are able to align the feature representations from the encoder with those from the regressor to be close to the same feature space of CLIP latents. In this way, both encoder and regressor can be fully learned, especially on the encoder that is transferred for downstream tasks. The two latent alignment serve as loss functions, in which they are robust to different loss types and the cosine distance is used by default.

We pretrain CAE v2 on ImageNet-1K images without using specific ground-truth labels. Extensive experiments demonstrate that our CAE v2 achieves competitive results across all scales of models, from the tiny

size to the large size, on various downstream tasks, including image classification, semantic segmentation, object detection, and instance segmentation.

In summary, our contributions are:

- We develop a new MIM pipeline, CAE v2, which is an improved version of CAE with CLIP latent as the pretraining target for learning a high-quality encoder and facilitating model convergence.
- We introduce visible latent alignment and masked latent alignment, aligning feature representations generated from the encoder and the regressor with CLIP latents.
- Experiments show that CAE v2 effectively improves the representative ability of the encoder, achieving competitive performance across model sizes and various downstream vision tasks.

## 2 CAE v2

CAE v2 is an improved variant of CAE (Chen et al., 2023). Compared with CAE, CAE v2 only maintains the encoder and the regressor while discarding the decoder (Section 2.2). Meanwhile, CAE v2 applies two latent alignment objectives that are supervised by CLIP latents - feature representations extracted from the vision branch of CLIP (Section 2.3), which is different from CAE that uses one alignment loss and one reconstruction loss.

### 2.1 Preliminary: CAE

The network of CAE (Chen et al., 2023) is an encoder-regressor-decoder architecture, including two pretraining tasks: masked representation prediction and masked patch reconstruction. The key of CAE is to decouple learning of encoder from completing the pretraining tasks, and making predictions in the encoded representation space. We illustrate the computational graph of CAE in Figure 5 (b) in Appendix.

Let $\mathbf{x} \in \mathcal{D}$ denote an input image. CAE first embeds $\mathbf{x}$ into $N$ patches, which are then divided into two non-overlapped sets, *i.e.*, visible patches $\mathbf{X}_v$ and masked patches $\mathbf{X}_m$. Here, $N = |v| + |m|$. The mask ratio is $\gamma = |m|/N$. The **encoder** takes the visible patches as input and outputs visible representations $\mathbf{Z}_v$; The **regressor** then predicts the latent representations of the masked patches $\mathbf{Z}_m$ conditioned on the positions of masked patches, which are expected to be aligned with the representations $\mathbf{Z}_m$ computed from the encoder; The **decoder** then reconstructs the masked patches $\mathbf{Y}_m$ from the predicted encoded representations $\mathbf{Z}_m$ to the form of target $\bar{\mathbf{Y}}_m$. CAE includes the **reconstruction loss** for the reconstruction of masked patches to the expected targets (*i.e.*, DALL-E), and the **alignment loss** for the alignment of features from the encoder and the regressor.

### 2.2 Architecture

The overview of the pipeline of CAE v2 is shown in Figure 1, and the computational graph is illustrated in Figure 5 (a) in Appendix. CAE v2 builds upon the foundation of CAE with two main structural modifications: i) discarding the decoder and ii) applying two alignment loss functions with CLIP latent as the target. Specifically, CAE v2 is an encoder-regressor architecture, in which visible latent alignment loss is directly applied on latent representations of visible patches computed from the encoder, and masked latent alignment loss acts on latent representations of masked patches predicted from the regressor. The supervision signals of these two losses are both CLIP latents.

**Encoder.** The encoder $\mathcal{F}$ only receives visible patches $\mathbf{X}_v$. It maps the visible patches $\mathbf{X}_v$ to the latent representations $\mathbf{Z}_v$ across a stack of transformer blocks that is based on self-attention. We employ a series of ViTs (Dosovitskiy et al., 2021) as the encoder, including ViT-Tiny, -Small, -Base and -Large.

**Regressor.** Following Chen et al. (2023), the latent contextual regressor $\mathcal{H}$ predicts the latent representations of masked patches $\mathbf{Z}_m$ from $\mathbf{Z}_v$ conditioned on the positions of masked patches. $\mathcal{H}$ performs as the same as cross-attention, in which the queries are learnable mask tokens $\mathbf{Q}_m$, and the keys and values are both the concatenation of $\mathbf{Z}_v$ and the output of previous layers ($\mathbf{Q}_m$ for the first layer).

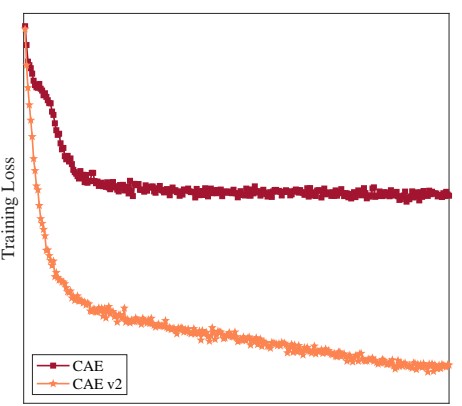 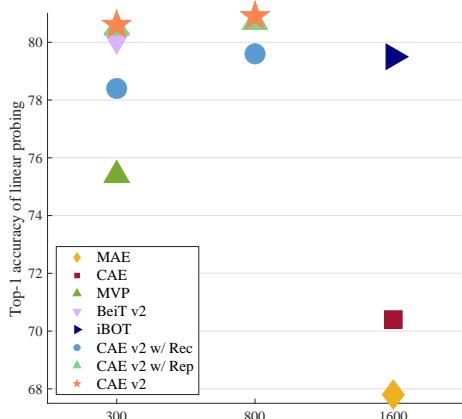

Figure 2: Analysis of the model convergence on (left) training loss and (right) top-1 accuracy of linear probing *vs.* epochs. The x-axis denotes the pretraining epoch.

**Decoder.** The decoder is to map masked latent representations $\mathbf{Z}_m$ to the format of target $\mathbf{Y}_m$. The decoder is based on a self-attention based transformer, different from the regressor based cross-attention. Different from CAE, CAE v2 does not have the decoder, since the regressor in CAE v2 can be responsible for predicting the latent representations of masked patches.

### 2.3 Objective Function

**Masking.** We adopt the random block-wise masking strategy as in BEiT (Bao et al., 2022). Instead of a unique mask ratio, we apply variable mask ratios for different sizes of models. In detail, we experimentally find that the optimal mask ratio is positively related with the model size, *i.e.*, the larger the model, the higher the mask ratio. More analyses are provided in Section 3.2.

**Targets.** We utilize the vision branch of CLIP (Radford et al., 2021) model to produce CLIP latent as the pretraining target. Specifically, the intact image is fed into the target model $\mathcal{T}$ to get latent representations of patches, which are then divided into the visible latent targets $\bar{\mathbf{Z}}_v$ and the masked latent targets $\bar{\mathbf{Z}}_m$ according to the position of visible and masked patches, respectively.

**Loss function.** CAE v2 consists of two loss functions: visible latent alignment and masked latent alignment. Visible latent alignment loss $\ell_v(\mathbf{Z}_v, \bar{\mathbf{Z}}_v)$ is applied on the latent representations of visible patches $\mathbf{Z}_v$ from the encoder, ensuring that the encoded representations lies in the latent representation space of the visible latent targets $\bar{\mathbf{Z}}_v$. Masked latent alignment loss $\ell_z(\mathbf{Z}_m, \bar{\mathbf{Z}}_m)$ allows the latent representations of masked patches $\mathbf{Z}_m$ predicted from the regressor to be close to the masked latent targets $\bar{\mathbf{Z}}_m$.

Overall, the whole loss function is:

$$\ell = \ell_v(\mathbf{Z}_v, \bar{\mathbf{Z}}_v) + \ell_z(\mathbf{Z}_m, \bar{\mathbf{Z}}_m), \tag{1}$$

By default, we use the cosine distance loss for both $\ell_v(\mathbf{Z}_v, \bar{\mathbf{Z}}_v)$ and $\ell_z(\mathbf{Z}_m, \bar{\mathbf{Z}}_m)$, although different kinds of loss types have negligible influences on CAE v2. In this way, the Eq. 1 can be refomulated as:

$$\ell = \ell_v(\mathbf{Z}_v, \bar{\mathbf{Z}}_v) + \ell_z(\mathbf{Z}_m, \bar{\mathbf{Z}}_m) = \frac{1}{|v|}\sum_{i=1}^{|v|}(1 - \cos(\mathbf{z}_v^i, \bar{\mathbf{z}}_v^i)) + \frac{1}{|m|}\sum_{i=1}^{|m|}(1 - \cos(\mathbf{z}_m^i, \bar{\mathbf{z}}_m^i)), \tag{2}$$

where $\mathbf{z}_v^i$ and $\bar{\mathbf{z}}_v^i$ represent the latent representation from the encoder of the $i$-th visible patch and its corresponding CLIP latent. Similarly, $\mathbf{z}_m^i$ and $\bar{\mathbf{z}}_m^i$ represent the latent representation from the regressor of the $i$-th masked patch and its corresponding CLIP latent. $\cos(\mathbf{u}, \mathbf{v}) = \frac{\mathbf{u} \cdot \mathbf{v}}{\|\mathbf{u}\|\|\mathbf{v}\|}$ represents the cosine similarity of two vectors.

**Differences between loss functions of CAE v2 and CAE (Chen et al., 2023).** CAE v2's loss functions (including visible latent alignment loss and masked latent alignment loss) differ from the original CAE's loss function (including reconstruction loss and alignment loss) on: i) *Target.* CAE v2's loss functions

both use CLIP latent as target, while the reconstruction loss of CAE uses DALL-E and the alignment loss uses latent representation from its encoder as target. ii) *Loss type.* CAE v2's loss functions both use the cosine distance loss by default, while CAE's reconstruction loss uses the cross-entropy loss and alignment loss uses the MSE loss. iii) *Mode.* CAE v2's loss functions directly optimize the encoder with visible latent alignment loss and the regressor with masked latent alignment loss, which encourages the encoder to be fully learned and fast convergence. Differently, CAE's reconstruction loss is applied on the decoder and alignment loss is used to align the representations from its regressor and its encoder, thus the encoder's learning is slow and implicit. These three differences on loss functions help our CAE v2 learn a better encoder than CAE. It can be verified on the linear probing downstream task that CAE v2 is superior than CAE by +10.3% (as shown in Table 6).

**Discussion on the positioning of CAE v2 with respect to CLIP (Radford et al., 2021).** CAE v2 is a MIM-based pre-training method, utilizing CLIP latent as the pre-training target. The reason of using CLIP latent is that it can provide rich semantics (Wei et al., 2022b) since CLIP is pre-trained on large-scale image-text pairs (*i.e.*, 400M image-pair private data). CAE v2 is dedicated to learning a high-quality visual encoder with only pre-training on images without any ground-truth labels. In doing so, the CLIP training data is only implicitly utilized via the pre-trained CLIP latents, while is not explicitly learned for training. It is different from CLIP that pre-trains on image-text pairs, therefore, CAE v2 is expected to perform better than CLIP on image-based downstream vision tasks. Experiments in Section 3.3 validate the accuracy improvements of CAE v2 with respect to CLIP. It is note that the CLIP vision model only extracts feature representations (*i.e.*, CLIP latents) as the pre-training targets, which is frozen during the pre-training of CAE v2. The encoder of CAE v2 is then transferred for the downstream task learning.

**Study and discussion.** Different from previous MIM works, CAE v2 directly optimizes the encoder with visible latent alignment loss $\ell_v$ on representations of the visible patches. Since supervision signals come from a semantically rich CLIP model, this loss can effectively facilitate the model convergence. It can be verified by Figure 2 that with 300-epoch pretraining schedule, the model with only using $\ell_v$ already achieves remarkable performance, which is largely superior than only using masked alignment loss $\ell_z$. When expending the pretraining schedule to 800 epoch, the gap between only using $\ell_v$ and only using $\ell_z$ reduces, showing that $\ell_v$ is good for the model convergence. Besides, the superior performance of only using $\ell_v$ on the linear probing task indicates that the representative ability of the encoder improves. The underlying reason is that $\ell_v$ successfully encourages the encoder to focus on the representation learning, instead of diverting some efforts to the masked patch prediction task. Consequently, the visible latent alignment plays an essential role in MIM pretraining task. Only using $\ell_z$ also achieves comparable performance. Even on the high performance of only using $\ell_v$, $\ell_z$ can further improves the performance, showing that aligning the encoder and the decoder to the same feature space is also effective for the encoder's learning.

## 3 Experiments

### 3.1 Settings

**Baseline.** The baseline of CAE v2 is replacing CAE's target to CLIP latent (as shown in No.1 in Table 1).

**Model structures.** We study a series of vision transformer backbones (Dosovitskiy et al., 2021), including ViT-Tiny (12 layers with $dim$=192), ViT-Small (12 layers with $dim$=384), ViT-Base (12 layers with $dim$=768), and ViT-Large (24 layers with $dim$=1024). Note that for ViT-Tiny, we follow Wang et al. (2022) to increase the number of heads from 3 to 12, which gives better results on ImageNet-1K (Deng et al., 2009). For other models, we strictly follow the model configurations as in Dosovitskiy et al. (2021).

For the target model, we adopt the vision branch CLIP-Base/16 of CLIP[1] for the pretraining experiments with ViT-Tiny/-Small/-Base and CLIP-Large/14 for ViT-Large. The size of input images is 224×224 for CLIP-Base/16 and 196×196 for CLIP-Large/14.

**Pretraining.** Following most previous MIM methods (Bao et al., 2022; He et al., 2022; Chen et al., 2023; Wei et al., 2022b; Wang et al., 2022), we use ImageNet-1K (IN-1K) dataset (Deng et al., 2009) for all pretraining experiments. The input images are with the size of $224 \times 224$ and are partitioned into $14 \times 14$

---

[1]The official pretrained CLIP model is available at `https://github.com/openai/CLIP/blob/main/clip/clip.py`.

Table 1: Individual change from CAE to CAE v2. $l_v$ and $l_z$ represent visible and masked latent alignment, respectively. The backbone is ViT-Base. All models are pre-trained for 300 epochs.

| No. | Method | IN-1K | | ADE20K | Object detection & instance segmentation | |
| | | LIN | FT | mIoU | Mask R-CNN | Cascade Mask R-CNN |
|---|---|---|---|---|---|---|
| 0 | CAE | 64.1 | 83.6 | 48.3 | 48.4 & 42.6 | 51.6 & 44.6 |
| 1 | + CLIP (baseline) | 75.8 | 83.6 | 49.5 | 49.3 & 43.0 | 51.7 & 44.6 |
| 2 | + $l_z$ | 78.4 | 85.0 | 52.7 | 51.8 & 44.3 | 53.8 & 45.8 |
| 3 | + $l_v$ (CAE v2) | **80.7** | **85.5** | **53.4** | **52.4** & **45.3** | **54.2** & **46.5** |

Table 2: Ablation studies for visible latent alignment loss $\ell_v$ and masked latent alignment loss $\ell_z$ in CAE v2. All models are pre-trained for 300 epochs. Default settings are marked in gray.

| Model | Loss function | | IN-1K | | ADE20K |
| | $\ell_z$ | $\ell_v$ | LIN | FT | mIoU |
|---|---|---|---|---|---|
| | ✓ | - | 64.9 | 77.2 | 44.1 |
| ViT-Tiny | - | ✓ | 68.8 | 77.4 | 44.2 |
| | ✓ | ✓ | **69.3** | **77.8** | **44.7** |
| | ✓ | - | 73.9 | 82.4 | 49.6 |
| ViT-Small | - | ✓ | 77.3 | 82.8 | 49.6 |
| | ✓ | ✓ | **77.5** | **83.1** | **49.8** |
| | ✓ | - | 78.4 | 85.0 | 52.7 |
| ViT-Base | - | ✓ | 80.5 | 85.2 | 53.1 |
| | ✓ | ✓ | **80.7** | **85.5** | **53.4** |

Table 3: Ablation studies for the type of loss function in CAE v2. Both $\ell_v$ and $\ell_z$ are used and all models are pre-trained for 300 epochs. We use the cosine distance by default (marked in gray).

| Model | Type of loss | IN-1K | | ADE20K |
| | | LIN | FT | mIoU |
|---|---|---|---|---|
| | MSE | 69.1 | 77.3 | **44.8** |
| ViT-Tiny | Smooth-$l1$ | **69.4** | 77.6 | 43.7 |
| | Cosine distance | 69.3 | **77.8** | 44.7 |
| | MSE | 77.3 | 82.7 | 49.8 |
| ViT-Small | Smooth-$l1$ | 77.4 | 82.8 | 49.8 |
| | Cosine distance | **77.5** | **83.1** | 49.8 |
| | MSE | 80.4 | 85.3 | 52.9 |
| ViT-Base | Smooth-$l1$ | 80.5 | 85.2 | 52.0 |
| | Cosine distance | **80.7** | **85.5** | **53.4** |

patches with the patch size being $16 \times 16$ across all sizes of models. We apply random resized cropping and horizontal flipping during pretraining.

The default mask ratios in the pretraining stage are set to 15%, 25%, 50%, and 60% on ViT-Tiny, -Small, -Base, and -Large, respectively. Without clear specification, we use AdamW (Loshchilov & Hutter, 2019) for optimization and train CAE v2 for 300 epochs across all scales of ViTs (Dosovitskiy et al., 2021). More detailed settings are listed in the appendix.

**Evaluation.** We evaluate our CAE v2 on various downstream tasks. For image classification, we conduct evaluations on ImageNet-1K (Deng et al., 2009) with both linear probing (LIN) and fine-tuning (FT) protocols. For semantic segmentation, we follow BEiT (Bao et al., 2022) to use UperNet (Xiao et al., 2018) and report the mIoU on ADE20K (Zhou et al., 2017) dataset. For objection detection and instance segmentation, we use COCO (Lin et al., 2014) as the evaluation dataset. We adopt both Mask R-CNN (He et al., 2017) and Cascade Mask R-CNN (Cai & Vasconcelos, 2018) frameworks and report $AP^b$ and $AP^m$ on the COCO val split. Please refer to the appendix for more training details on various downstream tasks.

### 3.2 Ablation Studies

In this subsection, we first illustrate individual change from CAE to CAE v2, and then analyse the effectiveness of visible latent alignment loss and masked latent alignment loss. After that we investigate mask ratio, type of loss, layer number of the regressor, and masking sampling strategy. Details are provided below.

**Individual change from CAE to CAE v2.** We experiment on each individual change from CAE to CAE v2 in Table 1. i) *CLIP latent as target.* We first change CAE's target from Dall-E to CLIP. Compared with CAE, this setting improves the performance, *e.g.*, +11.5%, 1.2% and 0.9% on ImageNet-1k's linear probing, ADE20K and COCO object detection, showing that CLIP latent as target can improve the representation ability of the encoder. This setting can be seen as our **baseline**. ii) *Masked latent alignment.* We then change the loss function to the proposed masked latent alignment. The performance is further improved by +2.6%, 1.4%, 3.2%, 2.5% on ImageNet-1k's linear probing, ImageNet-1k's fine-tuning, ADE20K, COCO object detection, respectively. It verifies the effectiveness of masked latent alignment. iii) *Visible latent alignment.* We further add the proposed visible latent alignment to directly optimize the encoder. The

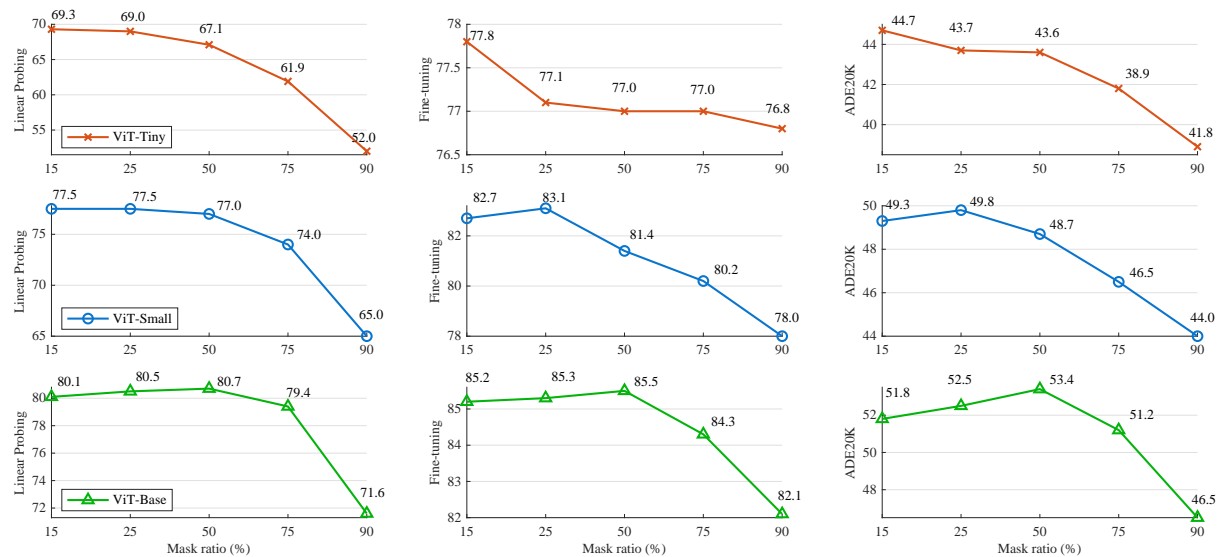

Figure 3: The optimal mask ratio is positive correlation with the model size in CAE v2. From top to bottom, the table shows the linear probing (left column), fine-tuning (middle column) on ImageNet-1K, and semantic segmentation (right column) on ADE20K with ViT-Tiny, -Small, and -Base model structure.

results are further increased by +2.3%, 0.5%, 0.7% and 0.6%, demonstrating the effectiveness of visible latent alignment.

**Visible latent alignment loss *vs.* masked latent alignment loss.** Previous MIM methods (Bao et al., 2022; He et al., 2022; Peng et al., 2022b; Liu et al., 2022b; Chen et al., 2023) typically apply the loss function on the predicted masked patches, as illustrated in Figure 5. Differently, CAE v2 applies two alignment losses on latent representations, *i.e.*, visible latent alignment loss $\ell_v$ on the representations of visible patches from the encoder, and masked latent alignment loss $\ell_z$ on the representations of predicted masked patches from the regressor. These two losses are independent that can be used either both of them or any individual one.

Table 2 shows the ablation results. One can see that the proposed $\ell_v$ and $\ell_y$ can steadily improve the performance compared to the baseline (*i.e.*, No.1 in Table 1). Specially, CAE v2 only with $\ell_y$ outperforms the baseline by large margins (*e.g.*, +2.6% on linear probing), showing the important role of masked latent alignment. Meanwhile, only using $\ell_v$ outperforms the strategy of only utilizing $\ell_z$, which verifies that the direct optimization on the encoder is beneficial for enhancing the encoder's representative quality. Combining $\ell_y$ with $\ell_v$ can achieve the best performance. Note that although the encoder is subject to the implicit influences from the masked patch prediction task to some extent, CAE v2 can still ensure a good optimization on the encoder, since the encoder directly receives supervision signals from the semantically rich CLIP model.

**Mask ratio.** We conduct experiments to analyse the influence of mask ratio, ranging from {15%, 25%, 50%, 75%, 90%}, across different scales of models. The results are listed in Figure 3. It shows that different scales of models prefer different values of mask ratios. The optimal mask ratio exhibits a positive correlation with the model size, *i.e.*, as the model size increases, a higher mask ratio performs better, and conversely, a smaller model benefits from a lower mask ratio. We also observe that when the mask ratio exceeds a stable value, the performances declines rapidly. The underlying reason might be that it is challenging for small-sized models to predict the masked patches from a limited subset of patches where most contextual information is missing. Therefore, using more visible patches can reduce the difficulty of the masked prediction task, benefiting for the model convergence. In contrast, large-scale models are easily subject to over-

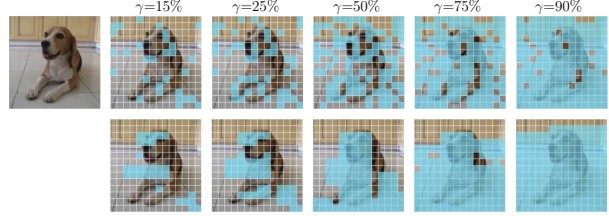

Figure 4: Illustration of corrupted images with different mask ratios $\gamma$ via the random sampling strategy (top row) and the block-wise sampling strategy (bottom row). We use the block-wise sampling strategy by default.

Table 4: The influence of the layer number of the regressor in CAE v2 and the decoder depth in MAE (He et al., 2022). Models are pre-trained for 300 epochs in CAE v2 and 800 epochs in MAE. Gray represents the default setting.

| Method | Model | # Layer | IN-1K | |
| --- | --- | --- | --- | --- |
| | | | LIN | FT |
| MAE | ViT-Large | 1 | 65.5 | 84.8 |
| | | 4 | 71.9 | 84.9 |
| | | 8 | 73.5 | 84.9 |
| CAE v2 | ViT-Base | 0 | 80.5 | 85.2 |
| | | 1 | 80.6 | 85.3 |
| | | 4 | **80.7** | **85.5** |
| | | 8 | 80.5 | 85.4 |

Table 5: Ablation studies for the mask sampling strategy in our CAE v2. All models are pre-trained for 300 epochs. We use the block-wise sampling by default (marked in gray).

| Model | Mask strategy | IN-1K | | ADE20K |
| --- | --- | --- | --- | --- |
| | | LIN | FT | mIoU |
| ViT-Tiny | Random | 69.1 | 77.4 | 43.8 |
| | Blockwise | **69.3** | **77.8** | **44.7** |
| ViT-Small | Random | 77.4 | 82.7 | 49.0 |
| | Blockwise | **77.5** | **83.1** | **49.8** |
| ViT-Base | Random | 80.6 | 85.4 | 52.4 |
| | Blockwise | **80.7** | **85.5** | **53.4** |

fitting when learning representations from overabundant visible patches, and thus a high mask ratio can make the masked prediction task more challenging to effectively mitigate the over-fitting problem. We set a mask ratio of 15%/25%/50%/60% for ViT-Tiny/-Small/-Base/-Large in our work unless specified.

**Type of loss.** We use cosine distance loss as illustrated in Eq. 2 for both visible and masked latent alignment loss functions. Instead of the cosine distance, it is possible to use other types of loss, like smooth-$\ell 1$ and mean square error (MSE), which are studied and utilized in the previous works (Peng et al., 2022a; Liu et al., 2022b; Wei et al., 2022c). We presents the results of these three kinds of loss types when using latent and masked alignment simultaneously in Table 3. The results show whatever the loss type is, our CAE v2 can achieve similar performance, while the cosine distance is slightly better than other types (*e.g.*, $\leq 0.5\%$ on linear probing and fine-tuning tasks). As previous works do (*e.g.*, MASKDISTILL (Peng et al., 2022b)), we choose the optimal loss type for CAE v2, *i.e.*, cosine distance by default.

**# layers in the regressor.** We study four choices, *i.e.*, {0-layer, 1-layer, 4-layer, 8-layer}, for the number of layers in the latent contextual regressor. The results in linear probing and fine-tuning in Table 4 show that the effect of the layer number is minor. In contrast, the decoder depth in MAE (He et al., 2022) has a large impact on downstream tasks, especially linear probing. We conjecture that our CAE v2 benefits from the explicit visible latent alignment on the visible patches, ensuring the representative quality of the encoder. When the regressor depth is 0, CAE v2 reduces to only using visible latent alignment loss, which still achieves satisfactory performance, indicating the effectiveness of this loss. We empirically use 4-layer regressor for ViT-Base/-Large and 1-layer regressor for ViT-Tiny/-Small by default.

**Mask sampling strategy.** We also compare different mask sampling strategies in our CAE v2, *i.e.*, the random sampling (He et al., 2022) and the block-wise sampling (Bao et al., 2022; Chen et al., 2023; Wei et al., 2022b) (as illustrated in Figure 4). Table 5 shows that there are only approximately 0.1% gaps between these two sampling strategies on linear probing and fine-tuning, but the block-wise sampling performs better than the random sampling on semantic segmentation (*e.g.*, 53.4% *vs.* 52.4% on ViT-Base). We use the block-wise sampling strategy as the default option.

### 3.3 Main Results

**Image classification on ImageNet-1K.** Table 6 shows comparisons of different models using two evaluation methods: linear probing and fine-tuning.

In linear probing on ImageNet-1K, CAE v2 demonstrates significant improvements over previous methods with other targets, *e.g.*, BEiT (Bao et al., 2022), MAE (He et al., 2022), CAE (Chen et al., 2023), and MaskFeat (Wei et al., 2022a). These gains are expected, as CLIP latents contain rich semantics than other targets. Compared to methods using CLIP latent as the target (MVP (Wei et al., 2022b) and MILAN (Hou et al., 2022)), CAE v2 also achieves superior performance (on ViT-Base with 300 epoch pretraining, CAE v2 *vs.* MVP: 80.7% *vs.* 75.4% and CAE v2 *vs.* MILAN: 80.7% *vs.* 78.9%).

In the fine-tuning task on ImageNet-1K, CAE v2 also achieves comparable performance across all scales of ViTs. Specifically, CAE v2 achieves **85.5%** top-1 accuracy with ViT-Base, surpassing all prior methods

Table 6: Pretraining evaluation on the top-1 accuracy (%) on linear probing (LIN) and fine-tuning (FT) on ImageNet-1K (Deng et al., 2009), and mIoU (%) on semantic segmentation on ADE20K (Zhou et al., 2017). ‡ means our implementation using the officially released code. § means the results from Chen et al. (2023) and Wei et al. (2022c). All other results except for ours are from the original papers. The methods achieving the best, second best and third place performance are denoted in **bold red**, green and blue respectively.

| Methods | #Epochs | Target | IN-1K | | ADE20K |
|---|---|---|---|---|---|
| | | | LIN | FT | mIoU |
| *Methods using ViT-Tiny*: | | | | | |
| MAE-Tiny (Wang et al., 2022) | 400 | RGB | 23.4 | 76.2 | - |
| CAE (Chen et al., 2023)‡ | 300 | DALL-E | 28.1 | 75.9 | 38.3 |
| Distilled MAE-lite (Wang et al., 2022) | 400 | RGB | - | 76.5 | - |
| G2SD (Wei et al., 2023) | 200 | MAE-Base | - | 77.0 | 44.5 |
| **CAE v2** | 300 | CLIP-Base | **69.3** | **77.8** | **44.7** |
| *Methods using ViT-Small*: | | | | | |
| MoCo v3 (Chen et al., 2021)§ | 300 | Self-EMA | 73.1 | 81.7 | - |
| BEiT (Bao et al., 2022)§ | 300 | DALL-E | 15.7 | 81.7 | - |
| SplitMask (El-Nouby et al., 2021) | 300 | DALL-E | - | 81.5 | - |
| CAE (Chen et al., 2023) | 300 | DALL-E | 51.8 | 82.0 | - |
| iBOT (Zhou et al., 2022a) | 3200 | Self-EMA | **77.9** | 82.3 | 45.4 |
| G2SD (Wei et al., 2023) | 200 | MAE-Base | - | 82.5 | 48.0 |
| **CAE v2** | 300 | CLIP-Base | 77.5 | **83.1** | **49.8** |
| *Methods using ViT-Base*: | | | | | |
| MoCo v3 (Chen et al., 2021) | 300 | Self-EMA | 76.5 | 83.2 | 47.2 |
| DINO (Caron et al., 2021)§ | 400 | Self-EMA | 77.3 | 83.3 | 47.2 |
| iBOT (Zhou et al., 2022a) | 1600 | Self-EMA | 79.5 | 84.0 | 50.0 |
| BEiT (Bao et al., 2022) | 800 | DALL-E | 56.7 | 83.2 | 45.6 |
| SimMIM (Xie et al., 2022) | 800 | RGB | 56.7 | 83.8 | - |
| MAE (He et al., 2022) | 1600 | RGB | 68.0 | 83.6 | 48.1 |
| CAE (Chen et al., 2023) | 1600 | DALL-E | 70.4 | 83.9 | 50.2 |
| SdAE (Chen et al., 2022) | 300 | Self-EMA | 64.9 | 84.1 | 48.6 |
| SIM (Tao et al., 2022) | 1600 | Self-EMA | 76.4 | 83.8 | - |
| MaskFeat (Wei et al., 2022a) | 1600 | HOG | - | 84.0 | - |
| SplitMask (El-Nouby et al., 2021) | 300 | DALL-E | - | 83.6 | 45.7 |
| PeCo (Dong et al., 2021) | 800 | VQGAN | - | 84.5 | 48.5 |
| data2vec (Baevski et al., 2022) | 800 | Self-EMA | - | 84.2 | - |
| CMAE (Huang et al., 2022b) | 1600 | RGB | - | 84.7 | 50.1 |
| ExtreMA (Wu et al., 2023) | 300 | Self-EMA | 73.3 | 83.7 | 47.9 |
| CLIP (Radford et al., 2021) | - | Text | 80.2 | 84.9 | 51.1 |
| MaskCLIP (Dong et al., 2023) | 1600 | Text | 72.9 | 84.1 | 50.8 |
| MVP (Wei et al., 2022b) | 300 | CLIP-Base | 75.4 | 84.4 | 52.4 |
| FD-CLIP (Wei et al., 2022c) | 300 | CLIP-Base | 80.3 | 84.9 | 52.8 |
| MILAN (Hou et al., 2022) | 400 | CLIP-Base | 78.9 | 85.4 | 52.7 |
| BEIT V2 (Peng et al., 2022a) | 1600 | VQ-CLIP-Base | - | 85.5 | 53.1 |
| dBOT (Liu et al., 2022b) | 1600 | CLIP-Base | – | **85.7** | 52.9 |
| MASKDISTILL (Peng et al., 2022b) | 300 | CLIP-Base | – | 85.0 | **53.8** |
| **CAE v2** | 300 | CLIP-Base | **80.7** | 85.5 | 53.4 |
| *Methods using ViT-Large*: | | | | | |
| MoCo v3 (Chen et al., 2021)§ | 300 | Self-EMA | - | 84.1 | 49.1 |
| BEiT (Bao et al., 2022)§ | 1600 | DALL-E | - | 85.2 | 53.3 |
| iBOT (Zhou et al., 2022a) | 1200 | Self-EMA | 81.0 | 84.8 | - |
| MAE (He et al., 2022) | 1600 | RGB | 75.8 | 85.9 | 53.6 |
| CAE (Chen et al., 2023) | 1600 | DALL-E | 78.1 | 86.3 | 54.7 |
| data2vec (Baevski et al., 2022) | 1600 | Self-EMA | - | 86.6 | - |
| CLIP (Radford et al., 2021)§ | - | Text | 83.5 | 86.1 | 53.5 |
| MVP (Wei et al., 2022b) | 300 | CLIP-Base | - | 86.3 | 54.3 |
| BEIT V2 (Peng et al., 2022a) | 1600 | VQ-CLIP-Base | - | 87.3 | 56.7 |
| FD-CLIP (Wei et al., 2022c) | 300 | CLIP-Large | **84.8** | 87.7 | 55.7 |
| MILAN (Hou et al., 2022) | 400 | CLIP-Large | 84.3 | **87.8** | **57.9** |
| dBOT (Liu et al., 2022b) | 1600 | CLIP-Large | – | **87.8** | 56.2 |
| MASKDISTILL (Peng et al., 2022b) | 300 | CLIP-Large | – | 87.6 | **57.9** |
| **CAE v2** | 300 | CLIP-Large | 84.4 | 87.6 | **57.9** |

Table 7: Pretraining evaluation on object detection (DET) and instance segmentation (INS) on COCO (Lin et al., 2014) with Mask R-CNN (He et al., 2017) (left) and Cascade Mask R-CNN (Cai & Vasconcelos, 2018) (right). All experiments are trained with the 1× schedule (12 epoch). Results except for CAE v2 are from Chen et al. (2023) and Liu et al. (2022b). #Epochs refers to the pretraining epochs on ImageNet-1K. * denotes multi-crop pretraining augmentation. The methods achieving the best and second best performance are denoted in **bold red** and green respectively.

| Method | #Epochs | Mask R-CNN | | Cascade Mask R-CNN | |
| | | DET | INS | DET | INS |
| | | $AP^b$ | $AP^m$ | $AP^b$ | $AP^m$ |
|---|---|---|---|---|---|
| *Methods using ViT-Small*: | | | | | |
| DeiT (Touvron et al., 2021) | 300 | 43.1 | 38.4 | - | - |
| MoCo v3* (Chen et al., 2021) | 300 | 43.3 | 38.8 | - | - |
| BEiT (Bao et al., 2022) | 300 | 35.6 | 32.6 | - | - |
| CAE (Chen et al., 2023) | 300 | 44.1 | 39.2 | - | - |
| iBOT* (Zhou et al., 2022a) | 3200 | - | - | 49.4 | 42.6 |
| **CAE v2** | 300 | **49.0** | **42.2** | **51.5** | **43.9** |
| *Methods using ViT-Base*: | | | | | |
| DeiT (Touvron et al., 2021) | 300 | 46.9 | 41.5 | - | - |
| MoCo v3* (Chen et al., 2021) | 300 | 45.5 | 40.5 | - | - |
| DINO* (Caron et al., 2021) | 400 | 46.8 | 41.5 | - | - |
| BEiT (Bao et al., 2022) | 800 | 42.1 | 37.8 | - | - |
| MAE (He et al., 2022) | 1600 | 48.4 | 42.6 | 51.3 | 44.3 |
| data2vec (Baevski et al., 2022) | 800 | 41.1 | 37.0 | - | - |
| iBoT (Zhou et al., 2022a) | 1600 | 48.6 | 43.1 | 51.2 | 44.2 |
| CAE (Chen et al., 2023) | 1600 | 50.0 | 44.0 | 52.9 | 45.5 |
| dBoT (Liu et al., 2022b) | 1600 | - | - | 53.6 | - |
| **CAE v2** | 300 | **52.4** | **45.3** | **54.2** | **46.5** |
| *Methods using ViT-Large*: | | | | | |
| MAE (He et al., 2022) | 1600 | 54.0 | 47.1 | - | - |
| data2vec (Baevski et al., 2022) | 1600 | 46.1 | 41.0 | - | - |
| iBoT (Zhou et al., 2022a) | 1600 | 50.6 | 44.7 | - | - |
| CAE (Chen et al., 2023) | 1600 | 54.5 | **47.6** | - | - |
| dBoT (Liu et al., 2022b) | 1600 | - | - | 56.8 | - |
| **CAE v2** | 300 | **55.2** | 47.3 | **56.9** | **48.6** |

except for dBOT (Liu et al., 2022b). Note that dBOT only outperforms CAE v2 by 0.2% while requiring 1600 epochs for pretraining, demonstrating CAE v2's higher efficiency.

**Semantic segmentation on ADE20K.** Semantic segmentation is a challenging task that needs to classify every pixel in an image according to various semantic labels. CLIP (Radford et al., 2021) latent serves as a powerful target for this task, showing clear advantages. As shown in Table 6, CAE v2 significantly outperforms methods pre-trained with other targets, *e.g.*, 3.2% mIoU improvement over CAE (Chen et al., 2023) when using ViT-Base. In comparison to MVP (Wei et al., 2022b) and BEIT V2 (Peng et al., 2022a), CAE v2 surpasses them with the same or less pretraining epochs. The superior performance persists when transitioning to ViT-Large, with which CAE v2 achieves **57.9%** mIoU on ADE20K (Zhou et al., 2017), outperforming previous methods.

**Object detection and instance segmentation on COCO.** We evaluate the pre-trained models on COCO (Lin et al., 2014) with Mask R-CNN (He et al., 2017) and Cascade Mask R-CNN (Cai & Vasconcelos, 2018; He et al., 2017) in Table 7. We report the results for 1× (12 epochs) training schedule. Compared with other pretraining methods, CAE v2 excels at both two configurations. With Mask R-CNN, CAE v2 achieves a 4.9% increase with ViT-Small and 2.4% increase with ViT-Base on $AP^b$ compared to previous best method Chen et al. (2023). The superior performance is maintained when employing Cascade Mask R-CNN as the fine-tuned model. For example, CAE v2 achieves a 0.6% increase with ViT-Base and 0.1% increase with ViT-Large on $AP^b$ compared to Liu et al. (2022b).

Compared with related works (*e.g.*, MASKDISTILL (Peng et al., 2022b) and MVP (Wei et al., 2022b)) that also utilize CLIP latent as target, the performance gap may be caused by: i) architecture choices. CAE v2 is built upon CAE (Chen et al., 2023) with structural modifications, while MASKDISTILL and MVP are based on BEiT Bao et al. (2022). ii) method. CAE v2 introduces visible and masked latent alignment to directly optimize the encoder and the regressor, respectively. Differently, MASKDISTILL and MVP supervise the overall model without separation. iii) loss type. Based on differences of the overall pipeline, different loss types perform inconsistent for achieving the best results. In detail, CAE v2 and MVP employ the cosine distance, while MASKDISTILL uses Smooth-$\ell_1$.

## 4 Related Work

Masked image modeling (MIM) aims to learn transferable vision representations for various downstream tasks. It is inspired by the successful large-scale pretraining for transformers (Vaswani et al., 2017) with masked language modeling (MLM) (Devlin et al., 2019; Chen et al., 2020a; Brown et al., 2020; Dong et al., 2019) in NLP and can serve as a pretext task in self-supervised vision pretraining (Caron et al., 2018; Doersch et al., 2015; Oord et al., 2018; Ermolov et al., 2021; Goyal et al., 2021; Li et al., 2021; Zbontar et al., 2021; He et al., 2020; Chen et al., 2020c;b; Grill et al., 2020; Liu et al., 2022a; Dong et al., 2022; Zhang et al., 2023; Huang et al., 2022a; Zhou et al., 2022b; Assran et al., 2022; Yi et al., 2023; Li et al., 2022). MIM methods (Bao et al., 2022; He et al., 2022; Xie et al., 2022; Chen et al., 2023; Baevski et al., 2022; Singh et al., 2022; Peng et al., 2022a; Fang et al., 2023b;a; Peng et al., 2022b; Liu et al., 2022b; Singh et al., 2023) follow a mask-then-predict pipeline of (i) corrupting an image by masking several image patches based on a pre-defined mask ratio and then (ii) learning to predict the missing content under specific targets as a reconstruction task. Our CAE v2 follows this pipeline while using more semantic targets and incorporating two alignment losses to effectively facilitate model convergence and improve the encoder's representative ability. In the following, we will discuss the related works.

**Targets.** Existing MIM methods explore different kinds of targets within their frameworks, including RGB pixels (He et al., 2022; Gao et al., 2022), hand-crafted features like HOG descriptors (Wei et al., 2022a), discrete visual tokens (Bao et al., 2022; Chen et al., 2023; El-Nouby et al., 2021; Peng et al., 2022a; Dong et al., 2021), and feature representations from momentum models (Tao et al., 2022; Chen et al., 2022; Wu et al., 2023) or from external pre-trained models (Wei et al., 2022a;b; Liu et al., 2022b; Peng et al., 2022b; Fang et al., 2023b;a; Gao et al., 2023; Ren et al., 2023). Among these methods, the features from the vision branch of CLIP model play a significant role as supervision signals, substantially improving the following downstream task performance (Wei et al., 2022b; Liu et al., 2022b; Peng et al., 2022b; Fang et al., 2023b;a; Gao et al., 2023; Ren et al., 2023; Hou et al., 2022; Ren et al., 2023). The underlying reason is that CLIP latents can provide multi-modality knowledge to aid vision-only MIM learning, which is beneficial for improving the ability of the encoder that is used for downstream tasks. In our CAE v2, we also use CLIP latents as the target for the model pretraining. Meanwhile, we go further one step to tap the potential of CLIP latent to further improve the quality of the encoder via the following introduced alignment losses.

**Loss function.** Most literature (Wei et al., 2022a; He et al., 2022; Chen et al., 2023; Bao et al., 2022; Peng et al., 2022a;b; Fang et al., 2023b;a; Ren et al., 2023; Liu et al., 2022b) apply the optimization to the predicted masked patches for the the masked patch prediction task. Following these methods, we also utilize a loss function for masked patch prediction task, which is called *masked latent alignment* since the optimization is applied to the representations of masked patches from the regressor. Furthermore, we find that directly applying supervision on the latent representations of visible patches from the encoder can constantly improve the model convergence and downstream transferring performance. We refer to this loss as *visible latent alignment*. Both visible and masked latent alignment use CLIP latent as the supervision target. In this way, the encoder can directly learn the semantic information from the powerful CLIP model, showing fast model convergence and high-quality encoder.

The most similar work to our approach is CAE (Chen et al., 2023). Our CAE v2 is an improved variant of CAE, yet there is no decoder. Meanwhile, both visible latent alignment and masked latent alignment are conducted on the latent representations that need to be similar to those from the feature space of CLIP model. Instead, CAE contains a decoder to recover the predicted masked representations to the discrete tokens, and align the predicted representations of the masked patches to the encoded representation space.

In addition, several concurrent works also attempt to explore the potential of CLIP latents (Wei et al., 2022b; Peng et al., 2022a;b; Fang et al., 2023b;a; Ren et al., 2023; Liu et al., 2022b). For example, MILAN (Hou et al., 2022) focuses on the architecture design, dBOT (Liu et al., 2022b) proposes a training strategy, and MASKDISTILL (Peng et al., 2022b) carefully studies optimization methods. FD-CLIP (Wei et al., 2022c) aims to improve the CLIP model with CLIP latent using feature distillation on intact image patches, which is not a MIM-based method. It is worth noting that our CAE v2 is orthogonal to these works. Specifically, we concentrate on tapping the potential of CLIP latent, and present CAE v2 with visible latent alignment and masked latent alignment that both use CLIP latent as the pretraining target, which is beneficial for learning a high-quality encoder and facilitating model convergence. The proposed CAE v2 demonstrates that the explicit visible latent alignment is beneficial for the representation learning of the encoder. Moreover, we experimentally explore the relationship between the mask ratio and the model size. We hope our findings can provide valuable guidelines for the future MIM pretraining.

## 5   Conclusion and Limitation

This paper introduces CAE v2, a context autoencoder with CLIP latent alignment, using the latent features from the vision branch of CLIP as the supervision target. CAE v2 consists of two alignment losses, *i.e.*, visible latent alignment loss - on the representations of visible patches from the encoder, and masked latent alignment loss - on the predicted representations of masked patches from the regressor. Extensive analyses and experiments show that our CAE v2 effectively facilities model convergence and achieves high performance on various downstream vision tasks. CAE v2 also outperforms the CLIP vision encoder on these vision tasks, although CAE v2 is not designed for direct comparisons with CLIP.

**Limitation.** Limited by resources, we do not examine larger models, such as ViT-Huge and ViT-Giant. We plan to explore this in the future and hope that the findings in CAE v2 will provide valuable guidance.

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

## A    Appendix

## B    Implementation Details

### B.1    Model structures

**Encoder.** For ViT-Tiny, we follow Wang et al. (2022) to increase the number of heads from 3 to 12. All others remain the same as the standard ViT architecture (Dosovitskiy et al., 2021). The end of encoder is a fully-connected layer (FC) followed by a layer normalization layer (LN) to map the target dimension to 512 for CLIP-Base and 768 for CLIP-Large. Note that FC and LN are discarded during the fine-tuning on downstream tasks.

**Regressor.** The regressor in CAE v2 is a stack of cross-attention based transformer blocks. For ViT-Tiny/-Small/-Base/-Large, we set the depth of regressor to 1/1/4/4, and the width to 96/384/768/1024. We also add a FC-LN after the regressor to map the target dimension, and these two layers share the same parameters to those in the encoder.

**Targets.** The targets in CAE v2 are derived from the last layer of the vision branch of CLIP model (Radford et al., 2021)[2], *i.e.*, the output of the projection head in the CLIP model with the dimension being 512 for CLIP-Base and 768 for CLIP-Large.

### B.2    Pipeline Comparison

We compare the computational graphs for (a) our CAE v2, (b) CAE (Chen et al., 2023), (c) BEiT (Bao et al., 2022), and (d) MAE (He et al., 2022) in Figure 5. The key novelty of CAE v2 is visible latent alignment and masked latent alignment, which are directly optimize the feature space of the encoder and regressor to be close to CLIP latents.

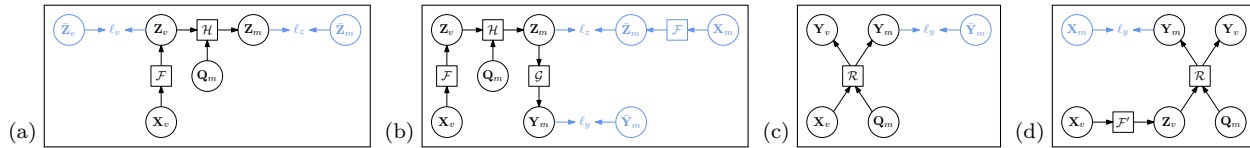

Figure 5: The computational graphs for (a) our CAE v2, (b) CAE (Chen et al., 2023), (c) BEiT (Bao et al., 2022), and (d) MAE (He et al., 2022). The parts in cornflower blue are for loss function. The encoder in (a) and (b) receives the visible patches and outputs their latent representations. The regressor predicts masked latent representations. (a) Visible latent alignment is applied on the representation learning of the encoder, and masked latent alignment loss is for predicting representations of masked patches. (b) The decoder reconstructs the masked patches from masked latent representations to discrete tokens. (c) The input includes both visible patches and mask queries, and the representations for both of them are updated within the function $\mathcal{R}$. (d) The encoder $\mathcal{F}'$ only processes the visible patches $\mathbf{X}_v$, and the decoder $\mathcal{R}$ inputs both latent representations and mask queries. For simplicity, the positional embeddings are not included in computational graphs.

### B.3    Training Details

**pretraining.** We present the default pretraining settings in Table 8. The pretraining epoch for all experiments is 300. For the CLIP model, we use base-size model for ViT-Tiny/-Small/-Base and the large-size model for ViT-Large. The input size is 224×224 for CLIP-Base and 196×196 for CLIP-Large. The input images are the same as those in the backbone model. We only use ImageNet-1K (Deng et al., 2009) for pretraining.

**Linear Probing on ImageNet-1K (Deng et al., 2009).** Following He et al. (2022); Doersch et al. (2015), we adopt an extra batch normalization layer (Ioffe & Szegedy, 2015) without affine transformation after the encoder. The default training details are shown in Table 9.

---

[2]The official pretrained CLIP model is available at `https://github.com/openai/CLIP/blob/main/clip/clip.py`.

Table 8: **Pretraining setting** for CAE v2 on ImageNet-1K.

| Config | Value |
|---|---|
| | ViT-Tiny/Small/Base/Large |
| Optimizer | AdamW (Loshchilov & Hutter, 2019) |
| Peak learning rate | 1.5e-3 |
| Minimal learning rate | 1e-5 |
| Optimizer momentum | $\beta_1, \beta_2 = 0.9, 0.98$ (Chen et al., 2020a) |
| Batch size | 2048 |
| Learning rate schedule | Cosine decay (Loshchilov & Hutter, 2017) |
| Warmup epochs (Goyal et al., 2017) | 10 |
| Max training epochs | 300 |
| Gradient Clipping | 3.0 |
| Weight decay | 0.05 |
| Drop path | 0.1/0.1/0.1/0.2 |
| Mask ratio | 0.15/0.25/0.5/0.6 |
| Mask strategy | Random block-wise sampling |
| Input scale | min=0.4, max=1.0 |
| Data Augmentation | Random resized crop & horizontal flip |
| Color jitter | 0.4 |
| Input size | 224×224 |

Table 9: **Linear probing setting** for CAE v2 on ImageNet-1K.

| Config | Value |
|---|---|
| | ViT-Tiny/Small/Base/Large |
| Optimizer | LARS (You et al., 2017) |
| Base learning rate | 0.1 |
| Weight decay | 0 |
| Optimizer momentum | 0.9 |
| Batch size | 16384 |
| Warmup epochs | 10 |
| Max training epochs | 90 |
| Data augmentation | Random resized crop & horizontal flip |

**Fine-tuning on ImageNet-1K (Deng et al., 2009).** The default settings for ImageNet-1K fine-tuning are in Table 10. The fine-tuning epoch for ViT-Tiny/-Small/-Base/-Large is 100/200/100/50 for fair comparison. The input size of all scales of models is 224×224. For ViT-Tiny, we sweep the learning rate ranging from 1e-3 to 4e-3. For ViT-Small/Base/Large, we select the learning rate from 1e-4 to 5e-4. By default, the learning rates and the layer-wise lr decay are 2e-3/5e-4/2e-4/4e-4 and 0.75/0.8/0.75/0.8 for ViT-Tiny/Small/Base/Large, respectively.

**Semantic segmentation in ADE20K (Zhou et al., 2017).** We follow the common setting in BEiT (Bao et al., 2022) to use UperNet (Xiao et al., 2018) as the task head and report the mIoU on ADE20K (Zhou et al., 2017). We add relative position bias (Raffel et al., 2020) during fine-tuning. For different scales of models in CAE v2, we search for the optimal learning rate and layer-wise learning rate decay in Table 11 for Table 6 in the main paper. Specifically, we select the optimal learning rate from {8e-5, 1e-4, 1.5e-4, 2e-4} for ViT-Tiny/Small/Base and from {1e-5, 2e-5, 3e-5, 4e-5} for ViT-Large.

**Object detection and instance segmentation in COCO (Lin et al., 2014).** We use COCO (Lin et al., 2014) for the evaluation on object detection and instance segmentation. We adopt both Mask R-CNN (He et al., 2017) and Cascade Mask R-CNN (Cai & Vasconcelos, 2018) frameworks and report $AP^b$ and $AP^m$ on the COCO val split. The input image is resized with the size of the short side between 480 and 800, while the size of the long side is no larger than 1333. Meanwhile, we use the relative position embedding and rotary position embedding (Su et al., 2021) during pretraining. Other training details are shown in Table 12.

Table 10: **Fine-tuning setting** for CAE v2 on ImageNet-1K.

| Config | Value |
|---|---|
| | ViT-Tiny/Small/Base/Large |
| Optimizer | AdamW (Loshchilov & Hutter, 2019) |
| Peak learning rate | {1, 2, 3, 4}e-3/ {1, 2, 3, 4, 5}e-4 |
| Minimal learning rate | 1e-6 |
| Weight decay | 0.05 |
| Optimizer momentum | $\beta_1, \beta_2$=0.9, 0.999 (Chen et al., 2020a) |
| Layer-wise lr decay (Bao et al., 2022; Clark et al., 2020) | {0.75, 0.8} |
| Batch size | 1024 |
| Learning rate schedule | Cosine decay (Loshchilov & Hutter, 2017) |
| Warmup epochs (Goyal et al., 2017) | 5 |
| Max training epochs | 100/100/100/50 |
| Data Augmentation | RandAug(10/9/9/9,0.5) |
| Label smoothing (Szegedy et al., 2016) | 0.0/0.1/0.1/0.1 |
| Mixup (Zhang et al., 2018) | 0.2/0.8/0.8/0.8 |
| Cutmix (Yun et al., 2019) | 0.0/1.0/1.0/1.0 |
| Color jitter | 0.3/0.4/0.4/0.4 |
| Drop path (Huang et al., 2016) | 0.0/0.1/0.1/0.2 |
| Input size | 224×224 |

Table 11: **Semantic segmentation setting** for CAE v2 on ADE20K.

| Config | Value |
|---|---|
| | ViT-Tiny/Small/Base/Large |
| Optimizer | AdamW (Loshchilov & Hutter, 2019) |
| Peak learning rate | {0.8, 1, 1.5, 2}e-4/{1, 2, 3, 4}e-5 |
| Minimal learning rate | 1e-6 |
| Weight decay | 0.05 |
| Optimizer momentum | $\beta_1, \beta_2$=0.9, 0.999 (Chen et al., 2020a) |
| Layer-wise lr decay (Bao et al., 2022; Clark et al., 2020) | {0.65, 0.75, 0.8, 0.85, 0.95} |
| Batch size | 16 |
| Learning rate schedule | Polynomial decay |
| Warmup steps | 1500 |
| Max training steps | 160000 |
| Drop path (Huang et al., 2016) | 0.1/0.1/0.1/0.15 |
| Input size | 512×512 |

Table 12: **Object detection and instance segmentation setting** for CAE v2 on COCO.

| Config | Value |
|---|---|
| | ViT-Small/Base/Large |
| Optimizer | AdamW (Loshchilov & Hutter, 2019) |
| Peak learning rate | {1, 1.5, 2, 3, 4}e-4 |
| Minimal learning rate | 1e-6 |
| Weight decay | 0.05 |
| Optimizer momentum | $\beta_1, \beta_2$=0.9, 0.999 (Chen et al., 2020a) |
| Layer-wise lr decay (Bao et al., 2022; Clark et al., 2020) | {0.8, 0.85, 0.95} |
| Batch size | 16 |
| Learning rate schedule | Step |
| Step epochs | 8, 11 |
| Max training epochs | 12 |
| Drop path (Huang et al., 2016) | 0.2 |

