# OpenReview forum: "CAE v2: Context Autoencoder with CLIP Latent Alignment"
_TMLR — Accepted by TMLR_

### Review · Reviewer_dQLm · 2023-06-21

**Summary Of Contributions:**

The paper proposes an improvement over CAE (called CAE-v2) where masked image modelling (MIM) is trained with CLIP latent vectors as targets in an encoder-regressor fashion. The results on multiple datasets and tasks (classification, segmentation and object detection with finetuning/linear probing) show improvement of this approach over the compared methods.

**Audience:**

Yes

**Claims And Evidence:**

Yes

**Requested Changes:**

see above

**Strengths And Weaknesses:**

## Strengths
1. The method is clearly explained comparing related approaches and overall the paper is well-written.
2. Extensive experiments conducted on various tasks and settings show consistent improvement of the proposed method over the compared approaches.

## Weaknesses
1. The benefit over CLIP is not clear. For masked and visible patches, CLIP latent is used as the target. Therefore, at best, the proposed model can be as good as CLIP (if I understand correctly). This raises the question that what is the purpose of this method and whether can it be better than CLIP. Please clarify this. For the second part of the question, I would like to see the CLIP method in comparisons wherever applicable.
2. Related to the above, if CAEv2 can be better than CLIP in some cases, I would like to see a justification why and results to support this.

---

> ### Author Response · Authors · 2023-07-23
> **Response to Reviewer dQLm**
>
> *The tables mentioned below are referred to our revised paper.*
>
> **Q1: The benefit of CLIP is not clear. For masked and visible patches, CLIP latent is used as the target. Therefore, at best, the proposed model can be as good as CLIP (if I understand correctly). This raises the question that what is the purpose of this method and whether can it be better than CLIP. Please clarify this. For the second part of the question, I would like to see the CLIP method in comparisons wherever applicable.**
>
> A1: Thanks for pointing out this issue.
>
> i) The purpose of CAE v2 is to use CLIP latent as target for learning a high-quality encoder and facilitating model convergence on our own task (*i.e.*, MIM-based pre-training on ImageNet-1K), rather than to outperform CLIP.
> CLIP is learned to align image and text representations from its own 400M image-text pair data.
> The differences in data and goals of CAE v2 and CLIP lead to differences in their ability to perform on different tasks.
>
> ii) Since CAE v2 is specialized on image pre-training task, it can achieve better performance than CLIP on some downstream tasks, like image classification and semantic segmentation.
> For example, CAE v2 outperforms CLIP by 7.4\%, 1.8\%, 2.2\% on linear probing, fine-tuning of ImageNet-1k and ADE20K (as illustrated in Table 6 in the revised paper).
>
> **Q2: Related to the above, if CAEv2 can be better than CLIP in some cases, I would like to see a justification why and results to support this.**
>
> A2:
> i) As illustrated in Q1, CAE v2 can be better than CLIP on some downstream tasks, like image classification and semantic segmentation.
> For example, CAE v2 achieves 80.7\%, 85.5\% and 53.4\% on linear probing, fine-tuning and ADE20k, which is superior than CLIP with 80.2\%, 84.9\% and 51.1\%.
>
> ii) The reason is that CAE v2 is designed for image pre-training task and focuses on learning high-quality vision representations from specific ImageNet-1K data, which is beneficial for these downstream tasks.
> In contrast, CLIP aims to learn the alignment between image and text, not specific for image pre-training.

---

### Review · Reviewer_rc6B · 2023-06-23

**Summary Of Contributions:**

The paper presents an improved masked image modeling (MIM) paradigm in which a vision encoder is learned by trying to align the encoded representations of both the visible and the masked targets with the correponding representations generated by a multimodal encoder like CLIP. Inspired from MVP, the approach modifies the CAE to use the representations from the CLIP visual encoder. The authors show that this simple adjustment leads to improved performance on several vision tasks — image classification, segmentation and detection.

**Audience:**

Yes

**Claims And Evidence:**

No

**Requested Changes:**

- I would like to see ablations performance on all tasks (classification, segmentation, detection) for a thorough evaluation. It seems like number of layers in regressors don't matter much for image classification? Does this trend hold true for segmentation or detection?
- The authors talk at several places about higher efficiency of their method. It would be nice if the paper includes training curves to back those claims.
- The authors might want to tone down the contribution claim about visual latent alignment and masked latent alignment. Current set of experiments and comparisons to existing baselines doesn't make it clear that the gains are not coming from other choices in the architecture (loss functions, etc).

**Strengths And Weaknesses:**

## Strengths

- Using the CLIP latents as targets for alignment makes sense (as shown by several other works like MVP). It is a natural step to try incorporate that into the CAE architecture.
- The approach greatly simplifies the training paradigm compared to previous works. Compared to CAE v1, it doesn’t need a decoder. Additionally, the model is able to distill  the rich representations learned by multi-modal encoders like CLIP to a vision only encoder.
- The experiments and ablations are exhaustive. Since the model architecture is simple, it has fewer moving pieces and all of them were ablated in the paper.

## Weaknesses
1. CLIP visual backbone matters much more than masked latent alignment.
- It doesn't seem like the performance varies a lot as the number of layers in the regressors are increased. As pointed out by the authors, with 0 layers in the regressor, the model only relies on the visible alignment loss. And this performs very close to vision + mask alignment loss at least for the image classification task. '
- This makes me wonder the role of masked latent alignment. Yes, it improves performance but only by a small margin. Most of the gains are coming from using a very rich visual backbone as the target (CLIP).

2. Comparisons to other approaches like CAE might be a bit unfair. The current approach uses CLIP which has been trained on a large amount of image-text data which other approaches like CAE doesn't have access to. This is also evident from the fact that methods that use CLIP as target have very similar numbers across all tasks.

3. The paper presents similar ideas as several recent works like MASKDISTILL, MVM, etc. I believe that the difference in performance is not due a fundamentally different idea, but subtle differences in architecture choices, losses (L1 vs Cosine, etc) etc. While the empirical improvements shouldn't be disregarded, I believe that the paper currently lacks in analyzing the performance gap with related works that explore the same idea of using a strong pre-trained visual encoder like CLIP as the target.


More clarifying questions:
- In Section 3.3, Semantic Segmentation on ADE20K, the authors say that CAE v2 surpasses CLIP with same or less pertaining epochs. Can you please clarify this line, given that CAE uses CLIP during pre-training?


## Final thoughts after reading the rebuttal:
I thank the authors for the detailed rebuttal. I went through the paper and the rebuttal again, and I am glad to see new experiments.
The new experiments addresses many of my concerns:
-- The authors show the contributions of each individual element in their approach (choice of loss function, masked latent alignment, visible latent alignment and demonstrate that each of them lead to improvement over baselines.
-- The authors also modified the original manuscript to explicitly list the differences with existing work (specially MVP and MASKDISTILL). I think that makes the manuscript much more clear, and as a reader, I definitely understand the contributions better now.
-- I am also happy to see the ablations table with other tasks as well. Thank you for providing such an exhaustive ablations table!

Overall, after reading the rebuttal and revised manuscript, I think my concerns have been adequately addressed and I recommend accepting the paper!

---

> ### Author Response · Authors · 2023-07-23
> **Response to Reviewer rc6B (1)**
>
> *The tables mentioned below are referred to our revised paper.*
>
> **Q1: CLIP visual backbone matters much more than masked latent alignment. It doesn't seem like the performance varies a lot as the number of layers in the regressors are increased. As pointed out by the authors, with 0 layers in the regressor, the model only relies on the visible alignment loss. And this performs very close to vision + mask alignment loss at least for the image classification task. This makes me wonder the role of masked latent alignment. Yes, it improves performance but only by a small margin. Most of the gains are coming from using a very rich visual backbone as the target (CLIP).**
>
> A1: Thanks for pointing out this.
>
> i) To clarify the role of masked latent alignment, we build an experiment without using both visible and masked latent alignment, i.e., merely replacing CAE's target to CLIP latent.
> The results on ViT-B are 75.8\%, 83.6\%, 49.5\% and 49.3\% on ImageNet-1k's linear probing, fine-tuning, ADE20K and COCO object detection, which are inferior than CAE v2 only with masked latent alignment with 78.4\%, 85.0\%, 52.7\% and 51.8\% (as illustrated in Table 1 in the revised paper), verifying the effectiveness of the proposed masked latent alignment.
>
> ii) Although CAE v2 with 0-layer regressor (i.e., CAE v2 only with visible latent alignment) achieves competitive performance, there is no doubt that masked latent alignment can still further improve the performance to a certain degree, e.g., +0.8\%, +0.4\% and +0.5\% with ViT-T, +0.2\%, +0.5\% and +0.4\% with ViT-B on fine-tuning, ADE20K and COCO object detection (as illustrated in Table 2 in the revised paper).
>
>
> **Q2: Comparisons to other approaches like CAE might be a bit unfair. The current approach uses CLIP which has been trained on a large amount of image-text data which other approaches like CAE doesn't have access to. This is also evident from the fact that methods that use CLIP as target have very similar numbers across all tasks.**
>
> A2: To make the comparisons to other approaches more fair, we modified the Table 6 and 7 to make separate comparisons according to whether using CLIP latent as target in the revised paper.
> Compared with methods using CLIP latent, our CAE v2 achieves competitive performance across all tasks and the state-of-the-art on object detection.
> For example, CAE v2 is superior than dBoT (Liu et al., 2022b) on COCO object detection and instance segmentation with cascade mask R-CNN (54.2\% *vs.* 53.6\%) with shorter pre-training process (300 epoch *vs.* 1600 epoch).
>
> **Q3: The paper presents similar ideas as several recent works like MASKDISTILL, MVM, etc. I believe that the difference in performance is not due a fundamentally different idea, but subtle differences in architecture choices, losses (L1 vs Cosine, etc) etc. While the empirical improvements shouldn't be disregarded, I believe that the paper currently lacks in analyzing the performance gap with related works that explore the same idea of using a strong pre-trained visual encoder like CLIP as the target.**
>
> A3: Thanks for your comments. The performance gap between CAE v2 with related works like MASKDISTILL, MVP, etc is caused by:
>
> * *Architecture choices*. CAE v2 is built upon CAE with structural modifications, while MASKDISTILL and MVP are based on BEiT.
> * *Method*. CAE v2 proposes visible and masked latent alignment to directly optimize the encoder and the regressor, respectively.
> Differently, MASKDISTILL and MVP supervise the overall model without separation.
> * *Loss type*. Based on differences of the overall pipeline, different loss types perform inconsistent for achieving the best results.
> In detail, CAE v2 and MVP employ the cosine distance, while MASKDISTILL uses Smooth-$\ell$1.
>
> We have have added above analyses on the performance gap in Section 3.3 in the revised paper.

---

> > ### Author Response · Authors · 2023-07-23
> > **Response to Reviewer rc6B (2)**
> >
> > **Q4: In Section 3.3, Semantic Segmentation on ADE20K, the authors say that CAE v2 surpasses CLIP with same or less pertaining epochs. Can you please clarify this line, given that CAE uses CLIP during pre-training?**
> >
> > A4: Thanks for pointing out this issue. We agree with you that using CLIP for comparison is not strictly appropriate since CAE v2 uses CLIP during pre-training. We have modified the description to "In comparison to MVP and BEIT V2 $\cdot\cdot\cdot$" in Section 3.3 in the revised version.
> >
> >
> > **Q5: I would like to see ablations performance on all tasks (classification, segmentation, detection) for a thorough evaluation. It seems like number of layers in regressors don't matter much for image classification? Does this trend hold true for segmentation or detection?**
> >
> > A5: We appreciate your insightful advice and have implemented an extensive ablation study to further probe the impact of the number of regressor layers across all tasks, including image classification, segmentation and detection, in the following table.
> >
> > The table shows that although the regressor depth has a minor effect on image classification, regressor with a relatively higher depth displays an enhanced performance especially for segmentation and detection tasks.
> > For example, \# Layer-4 outperforms \# Layer-0 by a margin of 0.5\% mIoU on the ADE20K dataset, and by 0.4\%/0.6\% AP on object detection/instance segmentation on COCO dataset respectively.
> > These results evidence the benefits of integrating masked latent alignment (\# Layer > 0) in order to improve performance on dense prediction tasks.
> > We also find that the performance gains plateau by further increasing the number of regressor layer. By default, we set the number of regressor layer to 4.
> >
> > **Q1A1's table. Ablation study on all tasks in terms of the layer number of the regressor in CAE v2 with ViT-Base as backbone.
> > Models are pre-trained for 300 epochs.**
> > | \# Layer  | LIN | FT  | ADE20K | Mask R-CNN  | Cascade Mask R-CNN |
> > | ------------- |:-------------|:-------------:|:-------------:|:-------------:|:-------------:|
> > | 0 | 80.5 | 85.2 | 52.9 | 52.0 \& 44.7 | 53.9 \& 45.8 |
> > | 1 | 80.6 | 85.3 | 52.9 | 52.0 \& 44.9 | 53.9 \& 45.9 |
> > | 4 | **80.7** | **85.5** | **53.4** | **52.4** \& **45.3** | **54.2** \& **46.5** |
> > | 8 | 80.5 | 85.4 | 52.7 | 52.2 \& 45.3 | 54.2 \& 46.2 |
> >
> >
> > **Q6: The authors talk at several places about the higher efficiency of their method. It would be nice if the paper includes training curves to back those claims.**
> >
> > A6: Thanks for your constructive advice. We have added the training curves of CAE and CAE v2 for comparison in Figure 2 in our revised version.
> >
> > **Q7: The authors might want to tone down the contribution claim about visual latent alignment and masked latent alignment. Current set of experiments and comparisons to existing baselines doesn't make it clear that the gains are not coming from other choices in the architecture (loss functions, etc).**
> >
> > A7: Thanks for your valuable suggestion.
> >
> > i) To clearly show the contribution of visual latent alignment and masked latent alignment, we added the studies in the following table.
> >
> > We build a baseline of CAE v2, *i.e.*, without both visual and masked latent alignment, which is equal to merely replace CAE's target to CLIP latent.
> > The table shows that both visible and masked latent alignment in CAE v2 can significantly improve the performance compared to the baseline.
> > In detail, masked latent alignment improves the baseline by +2.6\%, +1.4\%, +3.2\%, +2.5\% on linear probing, fine-tuning, ADE20K and COCO object detection, respectively.
> > Visible latent alignment can further improve the performance by +2.3\%, +0.5\%, +0.7\% and +0.6\% on linear probing, fine-tuning, ADE20K and COCO object detection, respectively.
> > | No.  | Method | LIN | FT  | ADE20K | Mask R-CNN  | Cascade Mask R-CNN |
> > | ------------- |:-------------|:-------------:|:-------------:|:-------------:|:-------------:|:-------------:|
> > | 0 | CAE | 64.1 | 83.6 | 48.3 | 48.4 \& 42.6 | 51.6 \& 44.6 |
> > | 1 | + CLIP (baseline) | 75.8 | 83.6 | 49.5 | 49.3 \& 43.0 | 51.7 \& 44.6 |
> > | 2 | + $l_z$ | 78.4 | 85.0 | 52.7 | 51.8 \& 44.3 | 53.8 \& 45.8 |
> > | 3 | + $l_v$ (CAE v2) | **80.7** | **85.5** | **53.4** | **52.4** \& **45.3** | **54.2** \& **46.5** |
> >
> > ii) We agree with you that the loss type of cosine distance achieves better results than others (*i.e.*, MSE and smooth-$l$1).
> > However, the performance gap caused by using different losses is minor (e.g., $\leq$0.5\% on linear probing and fine-tuning as shwon in Table 3 in the revised paper), especially compared with the performance improvement brought by the proposed visible and masked latent alignment (as shown in response i)).
> >
> > iii) We believe that the studies as described in response i) could highlight the contribution of visible and masked latent alignment more clearly, which have been added in Section 3.2 in the revised paper.

---

### Review · Reviewer_tWks · 2023-06-26

**Summary Of Contributions:**

The manuscript proposes a representation learning method that learns an image encoder from signals given by a pretrained CLIP image encoder. The method is similar to masked image models (e.g., masked autoencoders; MAE), which predicts the randomly-masked-out patch given the visible patches. However, instead of predicting raw patch pixels as in MAE, the method trains the encoder so that the encoding of visible patches is predictable of the CLIP encoding of *all* patches. The resulting method achieves higher performances on downstream tasks in less epochs, against both the base CLIP model and other baselines.

**Audience:**

Yes

**Claims And Evidence:**

Yes

**Requested Changes:**

1. Figure 3 and Table 1 show that L_m (the masked prediction loss that predicts the latents of masked patches) doesn't make much difference. Removing L_m leads to nearly identical performance. Admittedly, adding L_m increases performance by <0.5%, but this can be easily compensated by the ability to train longer because each iteration takes less compute. Longer training does improve performance according to Figure 3. If we care about total compute, we can train longer w/o L_m and get good performance. If we do not care, we can also just train as long as we want, and get good performance.Therefore, the introduction of L_m seems unnecessary to me.

2. Section 2.1 talks about the preliminary on CAE, the prior work this paper is based on. It would be great to expand this section in a bit more details, e.g., with notations to make it clearer.

3. Figure 2 talks about the computation graph of the proposed method and other baselines. The caption is really verbose and in my opinion the diagrams are not informative. I would suggest moving it to appendix, as Figure 1 is already quite clear..

**Strengths And Weaknesses:**

Strengths:
+ The method is simple and effective. The needed for fewer training epochs is nice.
+ Explanation of the method is generally clear, although some improvements can be made (see below).

Weaknesses:
+ The improvement is rather minor against baselines, even against the source model CLIP.
+ The experiments show that the masked-prediction loss may not be necessary.

---

> ### Author Response · Authors · 2023-07-23
> **Response to Reviewer tWks**
>
> *The tables mentioned below are referred to our revised paper.*
>
> **Q1: The improvement is rather minor against baselines, even against the source model CLIP.**
>
> A1: Compared with CAE v2 and CAE, CAE v2 surpasses CAE by +10.3\%, +1.6\%, 3.2\% and 2.4\% on downstream tasks of linear probing, fine-tuning, ADE20K and COCO object detection with ViT-B as shown in Table 6 and 7.
> Compared with CAE v2 and CAE only changing to CLIP target, CAE v2 has +2.6\%, 1.4\%, 3.2\%, 2.5\% improvement on these downstream tasks as illustrated in Table 1.
> We would like to emphasize that this improvement is perhaps more substantial than it might appear in the pre-training area.
>
> Compared with CAE v2 and CLIP, CAE v2 outperforms by 7.4\%, 1.8\%, 2.2\% on linear probing, fine-tuning, ADE20K (as shown in Table 6).
> Note that the goal of CAE v2 is *not* to outperform CLIP.
> Instead, the goal of CAE v2 is to use CLIP latent as target for learning a high-quality encoder and facilitating model convergence on ImageNet-1K. The superior performance over CLIP thus could be viewed as an ancillary benefit.
>
> **Q2: Figure 3 and Table 1 show that $L\_m$ (the masked prediction loss that predicts the latents of masked patches) doesn't make much difference. Removing $L\_m$ leads to nearly identical performance. Admittedly, adding $L_m$ increases performance by $<0.5\%$, but this can be easily compensated by the ability to train longer because each iteration takes less compute. Longer training does improve performance according to Figure 3. If we care about total compute, we can train longer w/o $L\_m$ and get good performance. If we do not care, we can also just train as long as we want, and get good performance.Therefore, the introduction of $L\_m$ seems unnecessary to me.**
>
> A2: Adding $L_m$ can consistently increase the performance, regardless of how long the training is.
> For example, $L_m$ boost the performance of linear probing from 80.5\% to 80.7\% on 300-epoch training, and from 80.7\% to 80.8\% improvement on 800-epoch training.
> We agree with you that the longer training can compensate the performance gap brought by $L_m$ in the short training process; however, it would take more than 2x longer training (from 300 epoch to 800 epoch).
> Taking both training efficiency and performance enhancement into account, $L_m$ proves its necessary to a certain degree.
>
> **Q3: Section 2.1 talks about the preliminary on CAE, the prior work this paper is based on. It would be great to expand this section in a bit more details, e.g., with notations to make it clearer.**
>
> A3: Thanks for your valuable suggestion. Following your suggestion, we have expanded Section 2.1 in more details to make it clearer, including notations.
>
> **Q4: Figure 2 talks about the computation graph of the proposed method and other baselines. The caption is really verbose and in my opinion the diagrams are not informative. I would suggest moving it to appendix, as Figure 1 is already quite clear.**
>
> A4: Thanks for your valuable advice. We have moved Figure 2 in the initial manuscript to appendix and simplified its caption following your suggestion.

---

### Review · Reviewer_4Cp8 · 2023-06-30

**Summary Of Contributions:**

Previous work introduced CAE, which utilizes CLIP latent representations as a target for masked image modeling, resulting in competitive performance across various downstream tasks. This paper introduces CAE v2, an updated version of CAE that replaces CAE’s regressor with a decoder and modifies the objective loss function.

**Audience:**

Yes

**Broader Impact Concerns:**

No concerns.

**Claims And Evidence:**

No

**Requested Changes:**

Address all the weaknesses that I commented in the previous box.

**Strengths And Weaknesses:**

Strengths:

- The paper presents comprehensive comparisons across multiple datasets.
- The method is presented in a clear and understandable manner.

Weaknesses:

- The paper lacks clarity in terms of its positioning with respect to CLIP. It is unclear whether the goal of CAE v2 is to outperform CLIP.  This clarification should be provided in the abstract and conclusions.
- The changes made to CAE are not adequately explained:
1. The distinction between a decoder and a regressor is unclear and requires further elaboration.
2. The modifications to the loss function are not clearly described. It is necessary to provide the mathematical expressions of the new loss functions and explain how they differ from the original CAE's loss function, highlighting the expected improvements.
3. Direct experimental comparisons between CAE and CAE v2 are ambiguous, especially when CAE utilizes DALE, making it difficult to assess the precise differences between the two approaches. The reported performance gap between CAE and CAE v2 is minimal in most experiments.
4. An ablation study should be conducted, systematically demonstrating the improvements achieved by each individual change from CAE to CAE v2.

In conclusion, while the paper demonstrates strengths such as extensive dataset comparisons and clear presentation, it would greatly benefit from addressing the weaknesses highlighted above, providing clarity on the relationship with CLIP, explaining the changes to CAE more thoroughly, and conducting a comprehensive ablation study to showcase the incremental improvements of each modification in CAE v2.

---

> ### Author Response · Authors · 2023-07-23
> **Response to Reviewer 4Cp8 (1)**
>
> **Q1: The paper lacks clarity in terms of its positioning with respect to CLIP. It is unclear whether the goal of CAE v2 is to outperform CLIP. This clarification should be provided in the abstract and conclusions.**
>
> A1: Thanks for pointing out this. The goal of CAE v2 is **not** to outperform CLIP. The goal of CAE v2 is to use CLIP latent as target for learning a high-quality encoder and for facilitating model convergence on our own task (*i.e.*, MIM-based pre-training on ImageNet-1K). We have added this clarification in the abstract and conclusions following your suggestion.
>
> **Q2: The distinction between a decoder and a regressor is unclear and requires further elaboration.**
>
> A2: Thanks for the comments. The distinction between a decoder and a regressor is:
> i) The regressor is used to predict the latent representations of masked patches, while the decoder is to map these latent representations to the format of target.
> ii) The regressor is a cross-attention based transformer,
> while the decoder is a self-attention based transformer.
> We have added this elaboration in Section 2.2 in the revised paper.
>
>
> **Q3: The modifications to the loss function are not clearly described. It is necessary to provide the mathematical expressions of the new loss functions and explain how they differ from the original CAE's loss function, highlighting the expected improvements.**
>
> A3: Thank you for the comments.
>
> i) The new loss functions (including visible latent alignment loss and masked latent alignment loss) differ from the original CAE's loss function (including reconstruction loss and alignment loss) on:
>
> * *Target*. New loss functions both use CLIP latent as target, while CAE's reconstruction loss uses DALL-E and alignment loss uses latent representation from its encoder as target.
> * *Loss type*. New loss functions both use the cosine distance loss, while CAE's reconstruction loss uses the cross-entropy loss and alignment loss uses the MSE loss.
> * *Manner*. New loss functions directly optimize the encoder with visible latent alignment loss and the regressor with masked latent alignment loss, which encourages the encoder to be fast and fully learned.
> In contrast, CAE's reconstruction loss is applied on the decoder and alignment loss is used to align the representations from its regressor and its encoder, thus the encoder's learning is slow and implicit.
>
> These three differences on loss functions help our CAE v2 learn a better encoder than CAE. It can be verified on the linear probing downstream task that CAE v2 is superior than CAE by +10.3\% (as shown in Table 6 in the revised paper).
>
> We have added the above explanation in Section 2.3 with "Differences between loss functions of CAE v2 and CAE." paragraph in our revised paper.
>
> ii) The mathematical expressions of the new loss functions (including visible latent alignment loss $\ell_v$ and masked latent alignment loss $\ell_z$) are:
>
> $$\ell_v(\mathbf{Z}\_v, \bar{\mathbf{Z}}\_v) + \ell_z(\mathbf{Z}\_m, \bar{\mathbf{Z}}\_m) = \frac{1}{\left | v \right |}\sum_{i=1}^{\left | v \right |}(1-\mathrm{cos}(\mathbf{z}^i\_v, \bar{\mathbf{z}}^i\_v)) + \frac{1}{\left | m \right |}\sum_{i=1}^{\left | m \right |}(1-\mathrm{cos}(\mathbf{z}^i\_m, \bar{\mathbf{z}}^i\_m)), $$
>
> where $\mathbf{z}^i\_v$ and $\bar{\mathbf{z}}^i\_v$ represent the latent representation from the encoder of the $i$-th visible patch and its corresponding CLIP latent.
> Similarly, $\mathbf{z}^i\_m$ and $\bar{\mathbf{z}}^i\_m$ represent the latent representation from the regressor of the $i$-th masked patch and its corresponding CLIP latent.
> $\mathrm{cos}(\mathbf{u}, \mathbf{v})=\frac{\mathbf{u}\cdot\mathbf{v}}{\|\mathbf{u}\|\|\mathbf{v}\|}$ represents the cosine similarity of two vectors.
>
> We have added the mathematical expressions of the new loss functions in Section 2.3 in the revised paper.

---

> > ### Author Response · Authors · 2023-07-23
> > **Response to Reviewer 4Cp8 (2)**
> >
> > **Q4: Direct experimental comparisons between CAE and CAE v2 are ambiguous, especially when CAE utilizes DALE, making it difficult to assess the precise differences between the two approaches. The reported performance gap between CAE and CAE v2 is minimal in most experiments.**
> >
> > A4: Thanks for pointing out this.
> >
> > To make the experimental comparisons between CAE and CAE v2 clear, we conduct an experiment to merely replace CAE's target to CLIP latent.
> > The result is 75.8\%, 83.6\%, 49.5\% and 49.3\% on ImageNet-1k's linear probing, ImageNet-1k's fine-tuning, ADE20K and COCO object detection with ViT-B, respectively, which is
> > better than CAE with 64.1\%, 83.6\%, 48.3\% and 48.4\%, yet still inferior than our CAE v2 with 80.7\%, 85.5\%, 53.4\% and 52.4\% (as illustrated in Table 1 in the revised paper).
> > It shows that CLIP latent and the proposed method both can boost the performance.
> >
> > CAE v2 outperforms CAE by +10.3\%, +1.6\%, 3.2\% and 2.4\% on linear probing, fine-tuning, ADE20K and COCO object detection with ViT-B as shown in Table 6 and 7.
> > We would like to emphasize that this improvement is perhaps more substantial than it might appear in the pre-training task.
> >
> > **Q5: An ablation study should be conducted, systematically demonstrating the improvements achieved by each individual change from CAE to CAE v2.**
> >
> > A5: Thanks for your valuable suggestion. We add experiments on each individual change from CAE to CAE v2 in the following table.
> >
> > * *CLIP latent as target*. We first change CAE's target from Dall-E to CLIP.
> > Compared with CAE, this setting improves the performance, *e.g.*, +11.5\%, 1.2\% and 0.9\% on ImageNet-1k's linear probing, ADE20K and COCO object detection, showing that CLIP latent as target can improve the representation ability of the encoder.
> > This setting can be seen as our baseline.
> >
> > * *Masked latent alignment*. We then change the loss function to the proposed masked latent alignment. The performance is further improved by +2.6\%, 1.4\%, 3.2\%, 2.5\% on ImageNet-1k's linear probing, ImageNet-1k's fine-tuning, ADE20K, COCO object detection, respectively.
> > It verifies the effectiveness of masked latent alignment.
> >
> > * *Visible latent alignment*. We further add the proposed visible latent alignment to directly optimize the encoder. The results are further increased by +2.3\%, 0.5\%, 0.7\% and 0.6\% on ImageNet-1k's linear probing, ImageNet-1k's fine-tuning, ADE20K, COCO object detection, demonstrating the effectiveness of visible latent alignment.
> >
> > The ablation study has been added in Table 1 in the revised paper.
> >
> > **Q5A5's table. Individual change from CAE to CAE v2. $l_v$ and $l_z$ represent visible and masked latent alignment. The backbone is ViT-Base. All models are pre-trained for 300 epochs.**
> > | No.  | Method | LIN | FT  | ADE20K | Mask R-CNN  | Cascade Mask R-CNN |
> > | ------------- |:-------------|:-------------:|:-------------:|:-------------:|:-------------:|:-------------:|
> > | 0 | CAE | 64.1 | 83.6 | 48.3 | 48.4 \& 42.6 | 51.6 \& 44.6 |
> > | 1 | + CLIP (baseline) | 75.8 | 83.6 | 49.5 | 49.3 \& 43.0 | 51.7 \& 44.6 |
> > | 2 | + $l_z$ | 78.4 | 85.0 | 52.7 | 51.8 \& 44.3 | 53.8 \& 45.8 |
> > | 3 | + $l_v$ (CAE v2) | **80.7** | **85.5** | **53.4** | **52.4** \& **45.3** | **54.2** \& **46.5** |

---

> > ### Comment · Reviewer_4Cp8 · 2023-08-14
> > **Quick clarification**
> >
> > Thanks for your answers. I think more clarity is needed in positioning the paper with respect to CLIP. It is not clear to state that CAE v2 goal is "to use CLIP latent as target for learning a high-quality encoder and for facilitating model convergence on our own task". This is because CLIP also has as a goal to be a high-quality encoder and facilitate model convergence to tasks. Can you please add further clarification how?

---

> > > ### Author Response · Authors · 2023-08-17
> > > **More clarity about the positioning of CAE v2 with respect to CLIP**
> > >
> > > Thanks for your response and suggestion.
> > > We agree that "more clarity is needed in the paper with respect to CLIP".
> > > We would like to provide more clarity about the positioning of CAE v2 with respect to CLIP as follows, and carefully revise the manuscript.
> > >
> > > i) Task. CAE v2 is an **image-based** MIM pre-training task, while CLIP is a **language-image** contrastive learning task.
> > >
> > > ii) Target signal.
> > > CAE v2 uses CLIP latent as target for pre-training, while CLIP uses the matched languages as target for images.
> > >
> > > iii) Goal. The goal of CAE v2 is to improve the encoder's quality and facilitate model convergence on **MIM pre-training task**.
> > > The utilization of CLIP latent as target is a method and strategy, since CLIP latent can provide rich semantics which is beneficial for achieving our goal as in [MVP, dBOT, BEIT V2, etc.].
> > > Differently, CLIP aims to improve the encoder's quality and facilitate model convergence on **downstream tasks**.
> > > Actually, CLIP uses large-scale datasets (i.e., 400M image-pair private data) during pre-training that is time-consuming.
> > >
> > > We conjecture that there might be a bit confusion about the term "our own task".
> > > We would like to clarify that this term points to the image-based MIM pre-training task, rather than downstream tasks.
> > >
> > > We will add the above clarification about the positioning of CAE v2 with CLIP in the revised version following your suggestions.

---

> > > > ### Comment · Reviewer_4Cp8 · 2023-08-18
> > > >
> > > > Thanks for your answer. This was already clear from the paper, I think the misunderstanding stems from the following question: Can you clarify how different are MIM pre-training task and downstream tasks it your opinion?

---

> > > > > ### Author Response · Authors · 2023-08-18
> > > > >
> > > > > Very thanks for your response and clarification of the misunderstanding.
> > > > >
> > > > > The main differences between MIM pre-training task and downstream task are:
> > > > >
> > > > > i) The MIM pre-training task is a self-supervised image-based representation learning method.
> > > > > It aims to obtain useful representations without the use of specific ground-truth labels like object classes.
> > > > > In detail, MIM pre-training task reconstructs masked image patches to a specific self-data format, e.g., RGB pixels, tokens from DALL-E, unsupervised features from CLIP.
> > > > >
> > > > > ii) Downstream tasks are performed after pre-training. They contribute to evaluate the quality of the pre-trained encoder from the pre-training stage.
> > > > > Different downstream tasks have different ground-truth objectives, such as image classification uses object classes, and object detection uses bounding boxes and object classes.
> > > > >
> > > > > We would like to further clarify that "high-quality encoder" should be evaluated by downstream tasks, and "facilitating model convergence" points to the pre-training stage in our CAE v2.

---

> > > > > > ### Comment · Reviewer_4Cp8 · 2023-08-18
> > > > > >
> > > > > > i) "It aims to obtain useful representations without the use of specific ground-truth labels like object classes." CAE needs pre-trained CLIP for training, and CLIP was trained with ground-truth labels. So, CAE does not fit into this definition of MIM pre-training because it uses ground-truth labels through the usage of CLIP.
> > > > > >
> > > > > > ii) "Downstream tasks are performed after pre-training.". So, CAE is not going to be used in downstream tasks? In which tasks is it going to be used then?

---

> > > > > > > ### Author Response · Authors · 2023-08-18
> > > > > > >
> > > > > > > Thanks for your response.
> > > > > > >
> > > > > > > i) We would like to clarify that CAE v2 is pre-trained on ImageNet-1K dataset. Only images, without labels, of the ImageNet-1K dataset are used for the pre-training. For CLIP latent, we also input the ImageNet-1K images into the CLIP vision encoder to obtain the feature representations of images. These extracted image features are used as target of CAE v2. In this case, CLIP also does not use ground-truth labels. In summary, on ImageNet-1K dataset, we do not use ground-truth labels.
> > > > > > >
> > > > > > > ii) CAE v2 is a pre-training method. After the pre-training, CAE v2 can obtain a pre-trained model. We use the encoder in the pre-trained model as the initialization weights and continue to fine-tuning on the downstream tasks. Thus, the pre-trained encoder of CAE v2 is also used on downstream tasks. In CAE v2, downstream tasks include image classification, object detection, and semantic segmentation. The pipeline, i.e., pre-training and then fine-tuning on downstream tasks, is a general and popular format in the current computer vision community. CAE v2 also follows this pipeline.
> > > > > > >
> > > > > > > Thanks again for your reply and we hope our response can well address your concern.

---

> > > > > > > > ### Comment · Reviewer_4Cp8 · 2023-08-19
> > > > > > > >
> > > > > > > > i) This confirms that the definition should be corrected to include that it targets exclusively not using ImageNet-1K labels, but other ground-truth labels are allowed given that CLIP pre-trained is used in CAE v2: CAEv2 aims to obtain useful representations without the use of **ImageNet-1K** specific ground-truth labels.
> > > > > > > >
> > > > > > > > This confirms that CAEv2 and CLIP are trained using the same data, as CAEv2 uses pre-trained CLIP representations for training.
> > > > > > > >
> > > > > > > > ii) This answer confirms that CAEv2 and CLIP have the same final goals.
> > > > > > > >
> > > > > > > > This circles back to the initial point that the positioning with respect to CLIP is not accurate and unclear in the current version of the paper. CAEv2 is distilling CLIP into another architecture with the aim of tackling same tasks as CLIP. So, accuracy improvements should be expected with respect to CLIP, otherwise it is unclear what is the point of CAE v2 as it uses same (more) training data and same applications as CLIP.
> > > > > > > >
> > > > > > > > In summary, the paper should state clearly the positioning with respect to CLIP and provide a fair comparison with CLIP (ie. control for number of parameters, data etc.)

---

> > > > > > > > > ### Author Response · Authors · 2023-08-19
> > > > > > > > >
> > > > > > > > > Thanks for your response.
> > > > > > > > >
> > > > > > > > > i) We agree with you and we will correct and clarify the definition of not using ImageNet-1K ground-truth labels.
> > > > > > > > >
> > > > > > > > > We agree with you that CAE v2 uses pre-trained CLIP representations for training. We also would like to further clarify that the CLIP vision model is not trained (i.e., the parameter is fixed) during the pre-training in CAE v2. The CLIP vision model is pre-trained by the original paper with its private 400 million (image, text) pairs collected from the internet. CAE v2 only use CLIP for inference to extract the feature representations of ImageNet-1K images.
> > > > > > > > >
> > > > > > > > > ii) We will state clearly the positioning with respect to CLIP in the revised version following your suggestion, including that there are accuracy improvements are expected with respect to CLIP on image-based downstream tasks. We greatly appreciate your valuable suggestion that are beneficial to improve the quality of our paper. Thanks again!

---

### Review · Reviewer_U2yY · 2023-07-02

**Summary Of Contributions:**

This work presents the CAE v2 algorithm, which optimizes the visual encoder to project some of image patches similarly to the latent representations from CLIP as well as to recover the embeddings of the missing image patches. Through extensive empirical evaluations, the authors show that this method yield competitive performance compared to other methods also using CLIP latents.

**Audience:**

Yes

**Broader Impact Concerns:**

no clear negative society impacts from this.

**Claims And Evidence:**

Yes

**Requested Changes:**

See the weakness above.

**Strengths And Weaknesses:**

Strengths:

The authors clearly described the method through the text and the paradigm figure. They also presented many empirical results showing that the model achieves strong performance. The paper also presented ablation studies showing the effects of the loss components and the loss formulations. CAE v2 is also more computationally efficient and achieves good performance compared to other methods requiring far more computations.

This method also has a potential application point that may be omitted by the authors. Can the trained encoder serve as a computation-cheaper replacement for the full CLIP encoder? How similar are the predicted latents compared to the ground truth latent from CLIP for both the visible and invisible patches? Can the predicted results be combined together to serve as the latent to be used in the downstream task just as how CLIP latent will be used in the downstream task?


Weaknesses:

The improvement from this method is really minor. The difference shown in the ablation studies is also small, making it unclear whether all the components or the loss formulation are critical. The authors need to show more studies supporting the importance of their designs.

There are also some methods that achieve very similar performance to CAE v2, but are not clearly introduced in the paper and differentiated from CAE v2, like FD-CLIP.

Are CAE v2 encoders better than the CLIP encoder? I only see one comparison in ViT-Base, what about other architectures? The authors need to highlight this comparison, since it is critical that this method outperforms the CLIP encoder to be useful.

---

> ### Author Response · Authors · 2023-07-23
> **Response to Reviewer U2yY**
>
> **Q1: This method also has a potential application point that may be omitted by the authors. Can the trained encoder serve as a computation-cheaper replacement for the full CLIP encoder? How similar are the predicted latents compared to the ground truth latent from CLIP for both the visible and invisible patches? Can the predicted results be combined together to serve as the latent to be used in the downstream task just as how CLIP latent will be used in the downstream task?**
>
> A1: This is a good point! Although we cannot conclude that our trained encoder can serve as a computation-cheaper replacement for the full CLIP vision encoder due to *the limited data (only ImageNet-1K)* used in CAE v2, it would be worth performing investigations in this direction.
>
> We follow the advice to measure the similarities between the representations of our CAE v2 and CLIP's vision encoder on ImageNet-1K, CIFAR100, and MNIST. The average similarities between the representations across the dataset are 92\%, 90\%, and 89\% on ImageNet-1K, CIFAR100, and MNIST. Given the high similarities, we try to compare the representations on the downstream classification task (our CAE v2 *vs.* CLIP's vision encoder in terms of top1: 66.8\% *vs.* 68.3\% on ImageNet-1K, 68.2\% *vs.* 68.9\% on CIFAR100, and 13.2\% *vs.* 52.1\% on MNIST), which is achieved by aligning image representation with text representations as done in CLIP. The results show that our CAE v2 is able to achieve reasonable performance but may not as generalizable as CLIP's vision encoder across different image domains.
>
> Thanks for the insightful comments. We will add more investigations in future works in this direction.
>
> **Q2: The improvement from this method is really minor. The difference shown in the ablation studies is also small, making it unclear whether all the components or the loss formulation are critical. The authors need to show more studies supporting the importance of their designs.**
>
> A2: To make it clear, we add the studies in Table 1 in our revised paper.
> We build the baseline, *i.e.*, without both visible and masked latent alignment, which is equal to merely replace CAE's target to CLIP latent.
> Table 1 shows that both visible and masked latent alignment in CAE v2 can significantly improve the performance compared to the baseline.
> In detail, masked latent alignment improves the baseline by +2.6\%, 1.4\%, 3.2\%, 2.5\% on ImageNet-1k's linear probing, ImageNet-1k's fine-tuning, ADE20K, COCO object detection, respectively.
> visible latent alignment can further improve the performance by +2.3\%, 0.5\%, 0.7\% and 0.6\% on ImageNet-1k's linear probing, ImageNet-1k's fine-tuning, ADE20K, COCO object detection, respectively.
>
> The ablations in Table 3 in our revised paper aim to indicate that our CAE v2 is robust to the types of loss functions (MSE, Smooth-$\ell$1, and cosine distance). They yield similar performances ($\leq$$0.5\%$ on linear probing and fine-tuning). We use the cosine distance by default according to the experimental results.
>
> **Q3: There are also some methods that achieve very similar performance to CAE v2, but are not clearly introduced in the paper and differentiated from CAE v2, like FD-CLIP.**
>
> A3: Thank you for pointing out the missed detailed discussions with FD-CLIP.
>
> The goal of CAE v2 is to *use* CLIP latent as target for learning a high-quality encoder and for facilitating model convergence in MIM pre-training, while FD-CLIP aims to surpass the CLIP model with CLIP latent that is not a MIM-based method. The proposed visible and masked latent alignment in CAE v2 is applied on visible and masked latent representations respectively, which is different from FD-CLIP adopts feature distillation on intact image patches.
>
> We have incorporated the above differentiation in Section. 4 of the revised paper.
>
>
> **Q4: Are CAE v2 encoders better than the CLIP encoder? I only see one comparison in ViT-Base, what about other architectures? The authors need to highlight this comparison, since it is critical that this method outperforms the CLIP encoder to be useful.**
>
> A4: Thanks for your valuable suggestion.
>
> * Our CAE v2 achieves better performance than CLIP in some downstream tasks, like image classification with linear probing and fine-tuning and semantic segmentation.
> For example, CAE v2 outperforms CLIP by 7.4\%, 1.8\%, 2.2\% on linear probing, fine-tuning, ADE20K with ViT-B.
>
> * Following your suggestion, we add performance comparison with CAE v2 and CLIP on ViT-Large in the following table.
> It shows that our CAE v2 maintains superior performance than CLIP.
> We do not compare ViT-T and ViT-S architectures, since they are not available in CLIP.
> |Model|Method|LIN|FT|ADE20K|
> |-|-|-|-|-|
> |ViT-B|CLIP|80.2|84.9|51.1|
> |ViT-B|CAE v2|80.7|85.5|53.4|
> |ViT-L|CLIP|83.5|86.1|53.5|
> |ViT-L|CAE v2|84.4|87.6|57.9|

---

### Author Response · Authors · 2023-07-23
**General Response**

We thank all the reviewers for their insightful and constructive comments. We are encouraged that reviewers generally recognize the strengths of our paper in:

* Method: simple and effective [tWks, rc6B], make sense [rc6B].
* Experiment: strong performance [U2yY, dQLm], computationally efficient [U2yY, tWks], comprehensive comparisons [4Cp8, rc6B].
* Presentation: clear and understandable [U2yY, 4Cp8, tWks, dQLm], well-written [dQLm].

Also, we thank all reviewers for their valuable and constructive suggestions, which help us a lot in improving our paper. We have run some new experiments and added illustrations, especially a clear baseline, to help address the concerns brought up by the reviewers.

We have provided point-wise responses below and updated the paper in a revised version following reviewers' suggestions. We hope our responses can clarify all reviewers' confusion and alleviate all concerns.

We thank all reviewers’ time again and we always ready to solve your concerns.

---

### Decision · Action_Editors · 2023-08-23

**Recommendation:** Accept with minor revision

**Comment:**

This paper was reviewed by 5 expert reviewers. Initially, the authors did not submit the rebuttal on time which led to 2 reviewers giving a reject final rating and 1 accept final rating. After the authors submitted the rebuttal (2 weeks late), 2 other reviewers were able to take the rebuttal into account and they both gave accept final ratings. Reviewers found that most of the concerns are satisfactorily resolved in the rebuttal, and the paper, albeit a bit incremental over the authors' prior work CAE, had solid and strong experiments and achieved state-of-the-art results. Besides, the improvements proposed in this paper, especially the masked latent alignment loss, are proven to be significant in the ablation studies. The editor refrained from submitting the decision earlier, to enable authors and reviewers to have more discussion. Finally, the editor believes that discussions have get conclusive, and that most reviewers recommend acceptance of the paper or only have minor concerns. Hence, the editor recommends acceptance of the paper. The authors should take into account further reviewer concerns during the discussion after the rebuttal, including the clarifications about the positioning of the paper w.r.t. CLIP, and submit a minor revision of the paper.

**Audience:**

This paper is of interest to the TMLR audience of multi-modal learning and unsupervised/self-supervised learning on visual data.

**Claims And Evidence:**

This paper has strong and extensive experiments over several datasets and achieved state-of-the-art performance. Initially reviewers asked for some additional experiments, these are supplied in the rebuttal and most reviewers are satisfied with the updated experiments.

---

> ### Author Response · Authors · 2023-09-21
> **Camera-ready submission and modification**
>
> We sincerely appreciate the editor and all reviewers' time and efforts in reviewing our paper. We deeply thank your valuable and constructive suggestions for improving our paper.
> Following the editor's suggestion, we have revised our manuscript by further incorporating the reviewer's concerns during the discussion after the rebuttal into camera-ready version, including the clarifications about the positioning of the paper with respect to CLIP [4Cp8].
>
> We thank the editor and all the reviewers again!